# Long-Context Linear System Identification

**Oğuz Kaan Yüksel**
EPFL
Lausanne, Switzerland

**Mathieu Even**
Inria – ENS
Paris, France

**Nicolas Flammarion**
EPFL
Lausanne, Switzerland

## Abstract

This paper addresses the problem of long-context linear system identification, where the state $x_t$ of a dynamical system at time $t$ depends linearly on previous states $x_s$ over a fixed context window of length $p$. We establish a sample complexity bound that matches the *i.i.d.* parametric rate up to logarithmic factors for a broad class of systems, extending previous works that considered only first-order dependencies. Our findings reveal a "learning-without-mixing" phenomenon, indicating that learning long-context linear autoregressive models is not hindered by slow mixing properties potentially associated with extended context windows. Additionally, we extend these results to *(i)* shared low-rank representations, where rank-regularized estimators improve rates with respect to dimensionality, and *(ii)* misspecified context lengths in strictly stable systems, where shorter contexts offer statistical advantages.

## 1 Introduction

System identification, which consists of estimating the parameters of a dynamical system from observations of its trajectories, is a fundamental problem in many fields such as econometrics, robotics, aeronautics, mechanical engineering, or reinforcement learning (Ljung, 1998; Gupta et al., 1976; Moerland et al., 2023). Recent theoretical advances focused on linear system identification, where observations are of the form:

$$x_t = A^\star x_{t-1} + \xi_t \,, \tag{1}$$

for $t \geqslant 1$, with initialization $x_0 \in \mathbb{R}^d$, noise $\xi_t \in \mathbb{R}^d$ and design matrix $A^\star \in \mathbb{R}^{d \times d}$. Linear system identification (Simpkins, 1999) has been thoroughly studied, with recent interest in sharp non-asymptotic rates (Simchowitz et al., 2018; Sarkar & Rakhlin, 2019; Faradonbeh et al., 2018; Jedra & Proutière, 2019). The existing analyses, however, focus solely on order-1 time dependency, in which the law of $x_t$ only depends on the previous state $x_{t-1}$. For order-$p$ time dependencies, the literature on non-asymptotic rates becomes surprisingly scarce, as existing techniques do not extend to $p > 1$.

We study this more general setting, where the state $x_t$ depends on previous states $x_s$ for $s$ in a context window of length $p \in \mathbb{N}^*$, i.e.,

$$x_t = \sum_{k=1}^{p} A_k^\star x_{t-k} + \xi_t \,, \tag{2}$$

for $t \geqslant p$, the initialization $x_0, \ldots, x_{p-1} \in \mathbb{R}^d$, noise $\xi_t \in \mathbb{R}^d$ and design matrices $A_1^\star, \ldots, A_p^\star \in \mathbb{R}^{d \times d}$. This classical $p^{\text{th}}$-order vector autoregression model (Box et al., 2015; Brockwell & Davis, 1991; Hamilton, 2020) is termed *long-context linear autoregressive model*. The term *linear* refers to the (noisy) linear relationship between iterates and *long-context* refers to the context length $p$. Recent advances in autoregressive models and architectures such as transformers (Vaswani et al., 2017; Dosovitskiy et al., 2021; El-Nouby et al., 2024) highlight the importance of long-context and its impact on learning. Developing a theoretical understanding of long-context linear autoregressive models is a necessary first step toward tackling these more complex architectures.

Motivated by empirical evidence that high-dimensional data may share some lower-dimensional representation (Bengio et al., 2013; Hospedales et al., 2022), several works additionally studied the

problem of learning matrices $A_k^\star$ under the assumption that they are of low-rank (Alquier et al., 2020; Basu et al., 2019), for order-1 autoregressive models. In the long-context setting, this problem is further motivated by the fact that if there exists a lower-dimensional representation of the autoregressive process, this translates into shared kernels for the matrices $A_k^\star$.

Finally, a key challenge in long-context autoregressive models is *misspecification*: the system might have an *unknown* context window $p$ as in Equation (2). $p$ may be arbitrarily large and unknown by the statistician. She may then specify a context length $p'$ that can be much smaller, thus yielding the following two fundamental questions: can useful structure still be learned under misspecified context lengths? And what advantages, if any, arise from model misspecification?

Our contributions in *long-context linear systems identification* are then threefold.

**(i)** We derive statistical rates on the recovery of matrices $A_k^\star$ in terms of Frobenius norm, which depends on the number of trajectories $N$ and their length $T$, on the dimension $d$ and the context length $p$. These rates reveal a "learning-without-mixing" phenomenon as they do not have a deflation in effective sample size due to the mixing time of the autoregressive process. This first contribution is an attempt to fill the gap in linear system identification for long context lengths.

**(ii)** We study statistical guarantees for learning the matrices $A_k^\star$ assuming that they are all of rank at most $r \ll d$. We prove that the statistical rate reduces, and that rank-regularized estimators adapt to the low-rank structure.

**(iii)** We study a scenario under which the model is *misspecified*. Fitting a linear model with context length $p' < p$ instead of $p$, we show that the first $p'$ matrices are still learned. More importantly, the sample complexity of learning these matrices depends only on the misspecified context length, indicating that misspecification may benefit the model statistically, not just computationally.

Finally, we confirm these statistical rates through experiments that verify the scaling laws predicted by problem parameters. Due to space constraints, these experiments are provided in Section E.

## 2 RELATED WORKS

In multivariate linear regression, one observes $\{(x_i, y_i)\}_{i=1}^N$ from the model $y_i = A^\star x_i + \xi_i$, where matrix $A^\star \in \mathbb{R}^{d \times d}$ and the sequences of noise $\xi_i$ and inputs $x_i$ are *i.i.d.*. The number of samples $N$ needs to scale at least as $d^2$ for a good estimation of $A^\star$ with ordinary least squares estimator (Hsu et al., 2012; Wainwright, 2019) in Frobenius norm—$\|A^\star - \hat{A}\|_F^2 \ll 1$. However, in many domains, data is sequential, violating the *i.i.d.* assumption. In such domains, classical non-*i.i.d.* formulations, such as vector autoregressive models or discrete-time linear dynamical systems (LDS), as seen in Equation (1), are often employed. Most works used to deal with the non-*i.i.d.*-ness of the data through mixing time arguments that fall short when the spectral radius of $A^\star$ reaches 1, leading to rates of the form $\|\hat{A} - A^\star\|_F^2 = \mathcal{O}(d^2/(n(1-\rho)))$ or $\|\hat{A} - A^\star\|_{op}^2 = \mathcal{O}(d/(n(1-\rho)))$ for some spectral quantity $1 - \rho$ related to the mixing time of the process. These rates apply to the OLS estimator (Faradonbeh et al., 2018) and online settings (Hardt et al., 2018; Even, 2023) alike.

Simchowitz et al. (2018); Sarkar & Rakhlin (2019) have developed excitation-based arguments to leverage mixing-time independent statistical bounds for the OLS estimator, while Hazan et al. (2017); Jain et al. (2021) respectively used spectral filtering and reverse experience replay in the online setting to obtain such bounds. The estimation of low-rank features has been studied by Basu et al. (2019); Alquier et al. (2020) via nuclear norm regularization. Finally, learning parameters of dynamical systems from $N$ trajectories of length $T$ has previously been considered by Tu et al. (2024) in a more general framework than Equation (1).

Layers of complexity can be added to the LDS described in Equation (1). Mania et al. (2022); Foster et al. (2020) considered non-linear dynamics, that write respectively as $x_{t+1} = A^\star \phi(x_t, u_t) + \xi_t$ and $x_{t+1} = f^\star(x_t) + \xi_t$, where in the former $A^\star$ is to be estimated and $\phi$ is a known non-linearity, while in the latter $f^\star$ is to be estimated. Kostic et al. (2022) recently provided a general framework using Koopman operators, to estimate the parameters of some general Markov chain. Giraud et al. (2015) considered time-varying systems, with arbitrary context lengths, while Bacchiocchi et al. (2024) studied autoregressive bandits. Ziemann & Tu (2022) provided a framework for learning non-parametric dynamical systems with "little mixing": as their rates are not hindered by slow

mixing after a burn-in time (that may itself depend on mixing properties). We refer the reader to Tsiamis et al. (2023) for a survey on recent advances on non-asymptotic system identification of LDS such as in Equation (1). Surprisingly, there does not seem to be much known about *long-context LDS* in Equation (2), the counterparts of LDS in Equation (1) with a context window $p > 1$.

## 3 PROBLEM SETTING

For a matrix $M \in \mathbb{R}^{d_1 \times d_2}$ with singular values $\sigma_1, \ldots, \sigma_{\min\{d_1, d_2\}}$, we denote its squared Frobenius norm as $\|M\|_F^2 = \sum_{(i,j)} M_{ij}^2 = \sum_\ell \sigma_\ell^2$, operator norm as $\|M\|_{\mathrm{op}} = \max_\ell |\sigma_\ell|$, and nuclear norm as $\|M\|_* = \sum_\ell |\sigma_\ell|$. $I_d$ and $0_d$ denotes the identity and the null $d \times d$ matrices, respectively. $\boldsymbol{A} = (A_1, \ldots, A_p)$ denotes a rectangular matrix of size $d \times pd$ where each $A_i$ is $d \times d$ block.

### 3.1 DATA GENERATION PROCESS

Let $d, p \in \mathbb{N}^*$ be the dimension of the state space and the context length, respectively. Consider the following linear autoregressive process:

$$\forall t > 0 : \quad x_t = \sum_{k=1}^p A_k^\star x_{t-k} + \xi_t \,, \tag{3}$$

where $x_s = 0$ for any $s \leqslant 0$ and the noise $\xi_t$ is independent of the $x_s, \xi_s$ for $s < t$. This is a particular instance of the general linear autoregressive model in Equation (2) with initial conditions $x_0, \ldots, x_{p-1}$ set to 0 and the independent noise structure. We assume sub-Gaussian noise:

**Assumption 3.1.** *For all $t$, the noise $\xi_t$ is centered and isotropic:*

$$\mathbb{E}[\xi_t] = 0 \,, \quad \mathbb{E}[\xi_t \xi_t^\top] = \sigma^2 I_d \,,$$

*and each coordinate of $\xi_t$ is independent and $c^2 \sigma^2$-sub-Gaussian (Wainwright, 2019, Chapter 2) for some $c \geqslant 1$:*

$$\forall i \in [d] \quad \|(\xi_t)_i\|_{\psi_2} \leqslant c^2 \sigma^2 \,, \quad \text{where} \quad \|x\|_{\psi_2} = \sup_{k \geqslant 1} k^{-1/2} \mathbb{E}\left[|x|^k\right]^{1/k} \,.$$

Let $\mathrm{AR}(\boldsymbol{A}^\star, \sigma^2)$ denote the law of the sequence defined in Equation (3) where $\boldsymbol{A}^\star$ denotes $(A_1^\star, \ldots, A_p^\star)$ for brevity. Given $N$ independent sequences of length $T > p$:

$$\left\{ x_t^{(n)}, n \in [N], t \in [T] \right\}, \quad \text{where} \quad (x_t^{(n)})_{t \in [T]} \overset{i.i.d.}{\sim} \mathrm{AR}(\boldsymbol{A}^\star, \sigma^2) \,,$$

the goal of long-context linear system identification is to estimate the matrices $A_k^\star, k \in [p]$.

Lastly, we assume a condition on the design matrices $A_k^\star, k \in [p]$ that amounts to an operator norm bound. First, we define the following linear operators for any matrix $\boldsymbol{A} \in \mathbb{R}^{d \times pd}$:

**Definition 3.2.** *Let $M_{\boldsymbol{A}} \in \mathbb{R}^{Td \times Td}$ be the block-matrix with block entries of size $d \times d$:*

$$M_{\boldsymbol{A}}^{(i,j)} = A_{i-j} \,, \quad \text{for all} \quad 1 \leqslant j < i \leqslant j + p \leqslant T \,, \quad \text{and} \quad M_{\boldsymbol{A}}^{(i,j)} = 0_d \,, \quad \text{otherwise} \,.$$

**Definition 3.3.** *Let $L_\star \in \mathbb{R}^{Td \times Td}$ be the block-matrix with block entries of size $d \times d$:*

$$L_\star^{(1,1)} = I_d \quad \text{and} \quad L_\star^{(i,1)} = \sum_{k=1}^{\max\{i-1,p\}} A_k^\star L_\star^{(i-k,1)} \ \ 1 < i \leqslant T \,,$$

$$L_\star^{(i,j)} = L_\star^{(i-j+1,1)} \ \text{for all} \ 1 \leqslant i \leqslant j \leqslant p \,, \quad \text{and} \quad L_\star^{(i,j)} = 0_d \ \text{otherwise} \,.$$

$M_{\boldsymbol{A}}$ executes predictions from the given data with $\boldsymbol{A}$ and $L_\star$ generates the data from the noise. That is, letting $(M_{\boldsymbol{A}})_{t\cdot}, (L_\star)_{t\cdot} : \mathbb{R}^d \times \mathbb{R}^{Td}$ be the $t^{\mathrm{th}}$ block-row of $M_{\boldsymbol{A}}$ and $L_\star$, respectively, we have:

$$(M_{\boldsymbol{A}})_{t\cdot} \begin{pmatrix} x_1^{(n)} \\ \vdots \\ x_T^{(n)} \end{pmatrix} = \sum_{k=1}^p A_k x_{t-k}^{(n)} \,, \quad (L_\star)_{t\cdot} \begin{pmatrix} \xi_1^{(n)} \\ \vdots \\ \xi_T^{(n)} \end{pmatrix} = x_t^{(n)} \,, \quad \text{with } x_s = 0 \text{ for } s \leqslant 0 \,.$$

Therefore, the operator norm of $M_{\boldsymbol{A}}$ is a measure of the worst-case growth of the predictions. Moreover, $M_{\boldsymbol{A}^\star}$ is linked to the data-generating operator $L_\star$:

$$L_\star = I_{Td} + M_{\boldsymbol{A}^\star} L_\star \quad \Longrightarrow \quad L_\star = (I_{Td} - M_{\boldsymbol{A}^\star})^{-1} = I_{Td} + \sum_{i=1}^{T-1} (M_{\boldsymbol{A}^\star})^i \,.$$

We assume the following conditions on the design matrices:

**Assumption 3.4.** *There exists a known constant $D \geqslant 1$ such that $\|M_{\boldsymbol{A}^\star}\|_{\mathrm{op}} \leqslant D$.*

Theorem 3.4 is not restrictive as $D$ is arbitrary and only needs to be an upper bound on $\|M_{\boldsymbol{A}^\star}\|_{\mathrm{op}}$. However, the *knowledge* of $D$ is necessary, as it is used to confine the estimator in Section 3.2.

As the operator $M_{\boldsymbol{A}^\star}$ is a derived object over the full trajectory, it is important to relate Theorem 3.4 to conditions on the design matrices $A_k^\star$. In Theorem 3.5 below, we provide two different assumptions on the design matrices that ensure the boundedness of the operator norm of $M_{\boldsymbol{A}^\star}$ with the same constant. Both conditions *imply* Theorem 3.4.

**Proposition 3.5.** *Theorem 3.4 holds if one of the following holds:*

$$\textbf{(i)} \quad \sum_{i=1}^{p} \|A_i^\star\|_{\mathrm{op}} \leqslant D \,, \qquad \textbf{(ii)} \quad \|\boldsymbol{A}^\star\|_{\mathrm{op}} \leqslant \frac{D}{\sqrt{p}} \,.$$

There is no direct assumption on $L_\star$; yet, our results depend on well-behavedness of $\kappa$, the condition number of $L_\star$, which is related to $\Gamma_t$ that appears in Simchowitz et al. (2018); Sarkar & Rakhlin (2019). $\kappa$ is related to the system stability, as explained in Section 5.

**Definition 3.6.** *Let $\kappa$ be the condition number of $L_\star$, i.e., $\kappa := \dfrac{\|L_\star\|_{\mathrm{op}}}{\sigma_{\min}(L_\star)}$.*

### 3.2 CONSTRAINED LEAST SQUARES

A natural estimator is the *Ordinary Least Square* (OLS), defined as any minimizer of the square loss:

$$\hat{\boldsymbol{A}}_{\mathrm{OLS}} \in \mathrm{argmin}_{\boldsymbol{A}}\, \mathcal{L}(\boldsymbol{A}) \,, \quad \text{where} \quad \mathcal{L}(\boldsymbol{A}) := \frac{1}{NT} \sum_{n=1}^{N} \sum_{t=p}^{T} \left\| x_t^{(n)} - \sum_{k=1}^{p} A_k x_{t-k}^{(n)} \right\|^2 \,. \tag{4}$$

The OLS estimator has been considered in previous works (Simchowitz et al., 2018; Alquier et al., 2020; Faradonbeh et al., 2018; Sarkar & Rakhlin, 2019), albeit in the $p = 1$ case. Most of these works provide estimation rates on $\|\hat{A} - A^\star\|_{\mathrm{op}}$ or $\|\hat{A} - A^\star\|_F$, for marginally stable systems, i.e., under the assumption that $\rho(A) \leqslant 1$ (Alquier et al., 2020; Simchowitz et al., 2018; Basu et al., 2019) and in the general case (Sarkar & Rakhlin, 2019).

Instead of directly considering the OLS estimator, we consider the empirical minimizer of the square loss under a restricted set of matrices $\boldsymbol{A}$ that have a bounded operator norm:

$$\hat{\boldsymbol{A}} \in \mathrm{argmin}_{\boldsymbol{A}=(A_1,\ldots,A_p)} \{\mathcal{L}(\boldsymbol{A}) \mid \|M_{\boldsymbol{A}}\|_{\mathrm{op}} \leqslant D\} \,. \tag{5}$$

Note that the set

$$\mathcal{A}(D) := \{\boldsymbol{A} = (A_1, \ldots, A_p) \mid \|M_{\boldsymbol{A}}\|_{\mathrm{op}} \leqslant D\} \,,$$

is bounded, closed and convex. Hence, the empirical minimizer of the square loss over $\mathcal{A}(D)$ can be computed with projected gradient descent (Duchi et al., 2008) or the Frank-Wolfe algorithm (Jaggi, 2013) as done for $\ell^1$ constrained optimization. To avoid projecting onto the set $\mathcal{A}(D)$, following Theorem 3.5, it is possible to restrict $\mathcal{A}(D)$ further into

$$\mathcal{A}(D)' := \left\{ \boldsymbol{A} \mid \sum_{i=1}^{p} \|A_i\|_{\mathrm{op}}^2 \leqslant D^2 \right\}, \quad \text{and} \quad \mathcal{A}(D)'' := \left\{ \boldsymbol{A} \mid \|\boldsymbol{A}\|_{\mathrm{op}} \leqslant \frac{D}{\sqrt{p}} \right\}.$$

in order to ensure a condition directly on design matrices. Then, the empirical minimizer of the square loss over $\mathcal{A}(D)'$ or $\mathcal{A}(D)''$ can again be computed via projected gradient descent or the Frank-Wolfe algorithm, with simplified projection steps.

Lastly, we briefly remark that the diameter constraint in Equation (5) can be removed, *i.e.*, $\mathcal{A}(D)$ replaced by $\mathcal{A}(\infty)$, under an additional assumption on $NT$. This is explained in detail in Section 5.

### 3.3 LOW-RANK ASSUMPTION

A common assumption in multi-task and meta-learning is that high-dimensional data often shares a representation in a smaller space (Bengio et al., 2013; Tripuraneni et al., 2021; Hospedales et al., 2022; Boursier et al., 2022; Collins et al., 2022; Yüksel et al., 2024) The following low-rank assumptions are crucial, as they significantly improve the statistical complexity of the problem.

**Assumption 3.7.** *For all $k \in [p]$, $\mathrm{rank}(A_k^\star) \leqslant r$.*

**Assumption 3.8.** *There exists an orthonormal matrix $P^\star \in \mathbb{R}^{r \times d}$ and matrices $B_1^\star, \ldots, B_p^\star \in \mathbb{R}^{d \times r}$ such that $A_k^\star = B_k^\star P^\star$ for all $k \in [p]$.*

Note that Theorem 3.8 is an instance of Theorem 3.7. The factorization $A_k^\star = Q^\star C_k^\star$ is another subcase of Theorem 3.7, but is not considered as it leads to iterates that directly lie in the subspace spanned by $Q^\star$ and hence $Q^\star$ can be learned by treating iterates $x_t^{(n)}$ as independent. In order to benefit from the low-rank structure, we consider the following regularized estimator:

$$\hat{A} \in \mathrm{argmin}_{A \in \mathcal{A}_r(D)} \mathcal{L}(A)\,, \quad \text{where} \quad \mathcal{A}_r(D) := \{A \in \mathcal{A}(D) \mid \forall k \in [p], \ \mathrm{rank}(A_k) \leqslant r\}\,. \quad (6)$$

### 3.4 MISSPECIFICATION

The context length of the generative autoregressive process might be unbounded, too large for an efficient estimation, or apriori unknown. In any case, practitioners still have to set a context length $p' \in \mathbb{N}^\star$ for the estimator, which might differ from the true $p$. In this scenario, we need an additional boundedness assumption that relates the first $p'$ matrices of the ground truth.

**Assumption 3.9.** *There exists a constant $D'$ such that*

$$\left\| \left( M_{A^\star} - M_{A_{1:p'}^\star} \right) L_\star \right\|_{\mathrm{op}} \leqslant D'\,, \quad \text{where} \quad A_{1:p'}^\star = (A_1^\star, \ldots, A_{p'}^\star, 0_d, \ldots, 0_d)\,.$$

Instead of the estimator defined in Equation (6), we consider the following misspecified estimator:

$$\hat{A} \in \mathrm{argmin}_{A \in \mathcal{A}_{r,p'}(D)} \mathcal{L}(A)\,, \quad \text{where} \quad \mathcal{A}_{r,p'}(D) := \{A \in \mathcal{A}_r(D) \mid \forall p' < k \leqslant p, A_k = 0_d\}\,. \quad (7)$$

Theorem 3.9 is a strong assumption as it requires that $L_\star$ is well-behaved regardless of the sequence length $T$. Consequently, the misspecification results are more stringent than other results and apply to a smaller class of systems that still includes strictly stable systems as discussed in Section 5.

## 4 LONG-CONTEXT LINEAR SYSTEM IDENTIFICATION

In this section, we present statistical rates for the recovery of the design matrices in terms of Frobenius norm. Since the matrices $A$ lie in $\mathbb{R}^{d \times pd}$, the number of variables is $pd^2$. In the *i.i.d.* setting, the rates of the form $\|\hat{A} - A^\star\|_F^2 = \mathcal{O}(pd^2/(NT))$ are expected. The following theorem extends this rate for long-context linear dynamical system identification:

**Theorem 4.1.** *Let Theorems 3.1 and 3.4 hold. Then, for any $0 < \delta < e^{-1}$, there exists a constant $C(\delta)$ such that the estimator $\hat{A}$ in Equation (5) verifies with probability $1 - \delta$:*

$$\left\| \hat{A} - A^\star \right\|_F^2 \leqslant C(\delta) D^2 \frac{pd^2}{N(T-p)} \mathrm{polylog}(\kappa, p, d, N, T)\,. \quad (8)$$

The constant $C(\delta)$ depends mildly on the sub-Gaussianity constant $c$ as described in Section C.5 and a sketch of proof is provided in Section A. The rate is numerically verified in Figure 1 in Section E. Theorem 4.1 exhibits several interesting features.

First, it shows that despite the temporal dependencies in the data, learning still occurs at a pace reminiscent of the *i.i.d.* setting, with a logarithmic term adjustment. This implies that the number of samples required to learn the system is approximately the same as in the *i.i.d.* setting, except for the logarithmic factor. Therefore, even though the data is sequential and only *i.i.d.* at the sequence level, the number of iterates $N(T-p)$ represents the *effective* data size.

Second, the rate in Equation (8) exhibits a linear dependency on the context length $p$ instead of a quadratic dependency. This is only due to the number of parameters to be estimated, which is $pd^2$ instead of $d^2$ and not a deflation in $T$ by a factor of $p$, which implies the context length does not affect the effective sample size. The additive factor in $T - p$ is due to the fact that first iterates do not depend on the full context length, and thus are not as informative as the later iterates. More detailed discussions of Theorem 4.1, in comparison with previous work, can be found in Section 5.

**Low-rank setting.** Next, we extend the results to the low-rank setting:

**Theorem 4.2.** *Let Theorems 3.1, 3.4 and 3.7 hold. Then, for any $0 < \delta < e^{-1}$, there exists a constant $C(\delta)$ such that the estimator $\hat{\boldsymbol{A}}$ in Equation* (6) *verifies with probability $1 - \delta$:*

$$\left\| \hat{\boldsymbol{A}} - \boldsymbol{A}^\star \right\|_F^2 \leqslant C(\delta) D^2 \frac{prd}{N(T-p)} \mathrm{polylog}(\kappa, p, d, r, N, T) \,.$$

The improved statistical rate depends on $rd$ instead of $d^2$. Note, however, that this estimator cannot be computed in polynomial time, since the underlying optimization problem involves a non-convex constraint on the rank of all $A_k$. Several heuristics exist to approximate this estimator. One approach is the Burer-Monteiro factorization (Burer & Monteiro, 2003; 2004), which involves parameterizing $A_k$ as $A_k = B_k C_k$ with $B_k \in \mathbb{R}^{d \times r}$ and $C_k \in \mathbb{R}^{r \times d}$. This method relaxes the constraint to a convex set but results in a non-convex function. Another approach is *hard-thresholding* algorithms, which use projected (stochastic) gradient descent on the non-convex constraint set (Blumensath & Davies, 2009; Foucart & Subramanian, 2019).

Perhaps the most intuitive approach is to use nuclear norm regularization, which is a convex relaxation of the rank constraint:

$$\hat{\boldsymbol{A}} \in \operatorname{argmin} \left\{ \mathcal{L}_\lambda(\boldsymbol{A}) \mid \boldsymbol{A} \in \mathcal{A}(D) \right\}, \quad \text{where} \quad \mathcal{L}_\lambda(\boldsymbol{A}) = \mathcal{L}(\boldsymbol{A}) + \lambda \|\boldsymbol{A}\|_{*,\mathrm{group}}, \qquad (9)$$

and $\|\boldsymbol{A}\|_{*,\mathrm{group}} = \sum_{k=1}^p \|A_k\|_*$ is the *group-nuclear norm*. We leave the analysis of the nuclear norm estimator for future work.

While the low-rank estimator cannot be computed easily, substituting the constraint $\forall k, \mathrm{rank}(A_k) \leqslant r$ with $\mathrm{rank}(\boldsymbol{A}) \leqslant r'$ enables a closed-form solution for the optimization problem (Bunea et al., 2011). However, the latter constraint effectively includes the former only when $r' \geqslant pr$, which would lead to suboptimal dependencies on the context length. These constraints are equivalent only if all $A_k$ matrices project onto the same space: i.e., $A_k = QB_k$ for some $Q \in \mathbb{R}^{d \times r}$ and $B_k \in \mathbb{R}^{r \times r}$.

**Misspecification.** Lastly, we study linear long-context autoregressive prediction models under misspecified context lengths and show that partial learning still occurs for misspecified models:

**Theorem 4.3.** *Let Theorems 3.1, 3.4, 3.7 and 3.9 hold. Then, for any $0 < \delta < e^{-1}$, there exists a constant $C(\delta)$ such that the estimator $\hat{\boldsymbol{A}}$ in Equation* (7) *verifies with probability $1 - \delta$:*

$$\left\| \hat{\boldsymbol{A}} - \boldsymbol{A}_{p'}^\star \right\|_F^2 \leqslant C(\delta) D^2 (D' + 1)^2 \frac{p'dr}{N(T-p)} \mathrm{polylog}(\kappa, p', d, r, N, T) \,.$$

For $r = d$, we recover Theorem 4.1 (full-rank setting) for misspecified context windows. The main improvement in that case of Theorem 4.3 over Theorem 4.1 is the dependency on $p'$ instead of $p$. In practice, $p$ can be much larger than $p'$ and even on the order of $T$. In such a setting, learning all matrices $A_k^\star$ becomes impossible if $N$ is not large enough and one does not take advantage of the length $T$ of the sequences. One can instead misspecify the student with a context length of $p' \ll p$ such that $NT \gg p'd^2$, so that the first $p'$ matrices are still learned.

Lastly, we briefly remark that Theorem 4.1 provides a rate for the case where $p < p'$. The latter case can be seen under a well-specified setting by rewriting the ground truth model as $\boldsymbol{A}^\star = (A_1^\star, \ldots, A_p^\star, 0_d, \ldots, 0_d)$ where the last $p' - p$ indices are padded with null matrices. Learning in such a case is then answered by Theorem 4.1 with a worsened rate that depends on $p'$.

## 5 DISCUSSION

We now discuss the rates obtained in Section 4 and compare them with previous results obtained for linear dynamical systems. In particular, we comment the "learning-without-mixing" phenomenon, introduced by Simchowitz et al. (2018) for the first-order linear dynamical systems.

**Adaptation of first-order techniques ($p = 1$) to the long-context setting.** Here, we explain why techniques developed in the $p = 1$ setting, in particular those of (Simchowitz et al., 2018; Sarkar & Rakhlin, 2019), do not work for the $p > 1$ setting and why, even if adapted, they would fail to achieve the desired sharp dependency on $p$.

Observe that the multi-step dynamics can be cast as a 1-step dynamic using *block companion matrices*. Let $X_t^{(n)} = (x_t^{(n),\top}, \dots, x_{t+p-1}^{(n),\top})^\top \in \mathbb{R}^{pd}$, $\Xi_t^{(n)} = (0, \dots, 0, (\xi_t^{(n)})^\top)^\top \in \mathbb{R}^{pd}$ and let $\mathcal{A}^\star \in \mathbb{R}^{pd \times pd}$ be the companion matrix associated to $\boldsymbol{A}^\star$:

$$
\mathcal{A}^\star = \begin{pmatrix}
0_d & I_d & \cdots & 0_d \\
\vdots & \ddots & \ddots & \vdots \\
0_d & \cdots & 0_d & I_d \\
A_p^\star & A_{p-1}^\star & \cdots & A_1^\star
\end{pmatrix}.
$$

We have the relation $X_{t+1}^{(n)} = \mathcal{A}^\star X_t^{(n)} + \Xi_t^{(n)}$, reducing the problem to the $p = 1$ case by increasing the dimension from $d$ to $pd$. First, brute-force adapting previous results to this case (e.g. Basu et al., 2019; Simchowitz et al., 2018; Sarkar & Rakhlin, 2019) is not possible since these works assume that the noise covariance of the additive noise added at each step ($\Xi_t^{(n)}$ here) is the identity matrix, or at least is positive definite. In our case, the noise covariance is the $pd \times pd$ block-diagonal matrix, with $p - 1$ blocks equal to $0_d$ and the last one to $I_d$. The covariance matrix is thus non-invertible, preventing the use of previous works.

In addition, arguments based on system excitation (e.g. Basu et al., 2019; Simchowitz et al., 2018) are bound to incur an additional dependence on $p$, on top of the factors expected due to the dimensionality of the problem. In particular, as seen in the small-ball martingales argument by Simchowitz et al. (2018, Section 2.3), evaluating quantities like $\|(\mathcal{A} - \mathcal{A}^\star)X_t^{(n)}\|^2$ for the $(k, \nu, q)$-block martingale small-ball assumption requires $k \geqslant p$ as $p$ represents the minimum number of steps for noise to propagate in every direction. Consequently, these analyses lead to a suboptimal $p$ dependency.

Moreover, adapting the techniques developed in the $p = 1$ setting (Sarkar & Rakhlin, 2019) which relies on explicit factorization of the OLS estimator is challenging. In the $p > 1$ case, the higher-order dynamics complicate the factorization, and the data matrix takes a Toeplitz form, which is more difficult to handle.

**Learning-without-mixing.** We explain why our rates exhibit "learning-without-mixing". We begin by defining "learning-with-mixing" and discussing the factors that influence the mixing time $\tau_{\mathrm{mix}}$. We then introduce the concept of "learning-without-mixing" as exemplified by Simchowitz et al. (2018) and show that our bounds exhibit similar properties.

Let $\tau_{\mathrm{mix}}$ be the mixing time of the Markov chain $(X_t^{(n)})_{t \geqslant 0}$. In the *i.i.d.* setting (for which $\tau_{\mathrm{mix}} = 1$), the OLS estimator obtains the optimal rate $\|\hat{\boldsymbol{A}}_{\mathrm{OLS}} - \boldsymbol{A}^\star\|_F^2 = \mathcal{O}(pd^2/NT)$, since $pd^2$ is the dimension of the inputs. With non-*i.i.d.* but Markovian data, a naive strategy would be to emulate *i.i.d.*-ness and take only a sample every $\tau_{\mathrm{mix}}$ steps of the trajectory to compute the OLS estimator, thus having data that are approximately *i.i.d.* while dividing the number of samples by $\tau_{\mathrm{mix}}$. This naive "learning-with-mixing" estimator would yield $\|\hat{\boldsymbol{A}}_{\mathrm{naive}} - \boldsymbol{A}^\star\|_F^2 = \tilde{\mathcal{O}}(\tau_{\mathrm{mix}}pd^2/NT)$, where the mixing time appears as a cost of non-*i.i.d.*-ness and $\tilde{\mathcal{O}}$ hides the logarithmic terms in problem parameters $p, d, N, T$.

In our case, two components contribute to the mixing time, $\tau_{\mathrm{mix}}$. The first component is related to the *stability* or the *excitability* of the system and scales as $1/(1 - \rho)$, where $\rho = \|M_{\boldsymbol{A}^\star}\|_{\mathrm{op}} < 1$. When $\rho \ll 1$, this component has no impact, while $\rho$ tends to 1, the system is less stable and the Markov chain mixes more slowly. The second component is directly related to the *context length* $p$ of the process. Regardless of the factor $1/(1 - \rho)$ above, the mixing time of our Markov chain is larger than $p$: since noise is added only in the last block in the recursion $X_{t+1}^{(n)} = \mathcal{A}^\star X_t^{(n)} + \Xi_t^{(n)}$, starting from a given state, $p$ iterations at least are needed to eventually forget this given state. The naive *learning-with-mixing* benchmark rate is thus $\|\hat{\boldsymbol{A}} - \boldsymbol{A}^\star\|_F^2 \leqslant \max\left(1/(1 - \rho), p\right)pd^2/NT$.

In contrast, a rate of convergence that exhibits "learning-without-mixing" is a rate of the form $\|\hat{\boldsymbol{A}} - \boldsymbol{A}^\star\|_F^2 \leqslant Cpd^2/NT$ where $C \ll \tau_{\mathrm{mix}}$. Such a rate means that the matrix $\boldsymbol{A}^\star$ is learned without

paying the cost of non-*i.i.d.*-ness. For instance, in the $p = 1$ case, the rate of Simchowitz et al. (2018) does not worsen as $\rho$ tends to 1—in fact, $\rho \to 1$ actually improves their rates.

The bound presented in Theorem 4.1 takes the form $\tilde{\mathcal{O}}(D^2pd^2/(N(T-p)))$. Importantly, the dependencies on the underlying Markov chain are only through $D$ and $\ln \kappa$, which do not have a direct dependency on the mixing time. The dependency on $D$ is merely an operator norm upper bound and does not diverge as the mixing time grows to $\infty$. Similarly, $\ln \kappa$ is logarithmic in $T$ for systems of interest, as we discuss below.

**System stability and** $\kappa$. We now explain the behavior of $\kappa$ defined in Theorem 3.6. First, by Theorem C.11, we have that $\sigma_{\min}(L_\star) \geqslant \frac{1}{D+1}$ and, thus, it is sufficient to upper bound

$$\zeta(T) := \sup_{i,j \in [T]} \|L_\star^{(i,j)}\|_{\mathrm{op}} \geqslant \sup_{i,j \in [T]} \frac{\|L_\star^{(i,j)}\|_F}{\sqrt{d}} \geqslant \frac{\|L_\star\|_F}{\sqrt{dT}} \geqslant \frac{\|L_\star\|_{\mathrm{op}}}{\sqrt{dT}}, \tag{10}$$

to control $\kappa$. Equation (10) implies that if the noise at step $i$ contributes to step $j$, as measured by $L_\star^{(i,j)}$, at a polynomial rate in $(j-i)$, then $\kappa$ grows at most polynomially in $T$. For such a $\kappa$, the resulting dependency on $T$ is of order $\ln T$ and mild. Instead, if it is exponential in $(j-i)$, then $\ln \kappa$ grows linearly in $T$ and the dependency on $T$ cancels out in the rate.

We use the quantity $\zeta(T)$ to define *strictly stable*, *marginally stable* and *explosive* systems:

**Definition 5.1.** *An LDS as defined in Equation* (3) *is called*

$$\begin{aligned} \textit{strictly stable if} \quad & \zeta(T) = \mathcal{O}(\rho^T) \quad \textit{for some} \quad \rho < 1\,, \\ \textit{marginally stable if} \quad & \zeta(T) = \mathcal{O}(T^k) \quad \textit{for some} \quad k \in \mathbb{N}\,, \\ \textit{explosive if} \quad & \zeta(T) = \mathcal{O}(\rho^T) \quad \textit{for some} \quad \rho > 1\,. \end{aligned}$$

Theorem 5.1 is similar to the notions of strictly stable, marginally stable and explosive systems considered in (Simchowitz et al., 2018; Sarkar & Rakhlin, 2019) for $p = 1$. Let $\rho(A^\star) := \lambda_{\max}(A^\star)$ be the spectral radius of $A^\star$ and $V \Lambda V^{-1}$ be the Jordan normal form of $A^\star$. Then,

$$\|L_\star^{(i,j)}\|_{\mathrm{op}} = \|(A^\star)^{j-i}\|_{\mathrm{op}} = \|V \Lambda^{j-1} V^{-1}\|_{\mathrm{op}} \leqslant \|V\|_{\mathrm{op}} \|\Lambda^{j-i}\|_{\mathrm{op}} \|V^{-1}\|_{\mathrm{op}}\,.$$

Note that $\|V\|_{\mathrm{op}}$ and $\|V^{-1}\|_{\mathrm{op}}$ are constants. For upper bounding $\|\Lambda^{j-i}\|_{\mathrm{op}}$, consider the Jordan blocks $\{\Lambda_k\}$ of $\Lambda$, associated with the eigenvalues $\lambda_k$ of $A^\star$. Then, $\|\Lambda^{j-i}\|_{\mathrm{op}} \leqslant \sup_k \|\Lambda_k^{j-i}\|_{\mathrm{op}}$ and

$$\begin{aligned} \|\Lambda_k^{j-i}\|_{\mathrm{op}} = \|(\lambda_k I_n + N_n)^{j-i}\|_{\mathrm{op}} &= \left\| \sum_{m=0}^{\max\{j-i,n-1\}} \lambda_k^m \binom{j-i}{m} N_n^m \right\|_{\mathrm{op}} \\ &\leqslant \sum_{m=0}^{\max\{j-i,n-1\}} \rho(A^\star)^{j-i} \binom{j-i}{m}\,, \end{aligned}$$

where $n$ is the block size for the Jordan block $\Lambda_k$. Note here that $n$ does not scale with $T$.

In particular, for *strictly stable* systems of Simchowitz et al. (2018); Sarkar & Rakhlin (2019) with $\rho < 1$, $\zeta(T) = \mathcal{O}(\rho^T)$. For *marginally stable* systems of Sarkar & Rakhlin (2019) with $\rho < 1 + \frac{\gamma}{T}$ with some constant $\gamma > 0$, $\rho(A^\star)^{j-i} \leqslant e^\gamma$ and $\zeta(T) = \mathcal{O}(T^k)$ for some fixed $k$ that depends on the largest Jordan block of $A^\star$. For *explosive* systems of Sarkar & Rakhlin (2019) with $\rho > 1$, $\zeta(T) = \mathcal{O}(\rho^T)$. Thus, Theorem 5.1 provides a general categorization of the systems based on the growth of $\zeta(T)$ in $p > 1$ case. Furthermore, our analysis yields sharp rates for *strictly stable* and *marginally stable* systems previously considered only in the $p = 1$ setting.

**Search space diameter** $D$. Our analysis is based on the assumption that the diameter $D$ of the search space is bounded and, hence, not directly applicable to the OLS estimator in Equation (4). However, Theorem C.7 in Section C.1 extends the results of Theorems 4.1 to 4.3 to minimizers without a constraint on the diameter of the search space. This extension does not change the rates but requires the additional assumption that $NT = \tilde{\Omega}(p'^2 dr)$.[1] In the case of Theorem 4.1, this

---

[1] We use the convention that $r = d, p' = p$ for Theorem 4.1.

corresponds to a result for the OLS estimator, but necessitating a number of samples quadratic in context length. Below, we comment on why the diameter restrictions is required when $NT \ll p'^2 dr$.

As mentioned earlier in comparison with (Simchowitz et al., 2018; Sarkar & Rakhlin, 2019), the simple OLS factorization in the $p = 1$ case does not generalize to the $p > 1$ and the data matrix has a Toeplitz structure that is more difficult to control. In order to deal with these issues, as explained in the sketch of proof in Section A, we rely on techniques from empirical process theory. These techniques are applied to quantify the probability of the event in Equation (13), which hold for any empirical risk minimizer of the square loss. This leads us to the study of the concentration of the martingales defined in Equation (15) around their predictable variation, which is a key step in our analysis. A uniform concentration is possible only if there is a uniform lower bound on the variations of the martingales, which can be achieved using a set of well-behaved matrices $\|M_{\boldsymbol{A}} - M_{\boldsymbol{A}^\star}\|_F / \|M_{\boldsymbol{A}} - M_{\boldsymbol{A}^\star}\|_{\mathrm{op}}$. In order to translate these conditions on the design matrices without additional dimensional dependencies, we introduce the operator norm constraint.

Lastly, it is possible to extend our analysis to unconstrained OLS by establishing a general coarse upper bound on the operator norm $\|M_{\hat{\boldsymbol{A}}}\|_{\mathrm{op}} \leqslant K$. This allows us to consider uniform lower bounds to matrices $\boldsymbol{A}$ with $\|M_{\boldsymbol{A}}\|_{\mathrm{op}} \leqslant K$, which lead to a rate for the OLS estimator in a similar manner.

**Upper bound on $D'$.** The misspecification result in Theorem 4.3 requires the additional assumption given in Theorem 3.9. In Theorem C.6, we show that a good upper bound on $D'$ is possible when $D < 1$, *i.e.*, the system is strictly stable, by using the bound $\|L_\star\|_{\mathrm{op}} \leqslant 1/(1 - D)$. However, misspecification results are not, a priori, applicable to marginally stable systems, which limits the practical applicability of our results. We leave the investigation of misspecification results for marginally stable systems for future work.

**Practical implications.** The rates obtained in Section 4 holds true if the empirical risk minimizer $\hat{\boldsymbol{A}}$ is replaced by any estimate $\tilde{\boldsymbol{A}}$ that verifies

$$\mathcal{L}(\tilde{\boldsymbol{A}}) \leqslant \mathcal{L}(\boldsymbol{A}^\star). \tag{11}$$

The training error for the ground truth $\boldsymbol{A}^\star$ concentrates around $\sigma^2 d$ with a rate of $\mathcal{O}(1/\sqrt{NT})$, which is identical to that in the *i.i.d.* setting. Hence, any algorithm that optimizes the training error below the threshold $\sigma^2 d$ achieve the rates in Section 4.

Moreover, in Section D, we show that the rates extend to approximate minimizers, i.e., any estimate $\tilde{\boldsymbol{A}}$ that verifies

$$\mathcal{L}(\tilde{\boldsymbol{A}}) \leqslant \mathcal{L}(\hat{\boldsymbol{A}}) + \epsilon_{\mathrm{tr}}, \quad \text{where} \quad \epsilon_{\mathrm{tr}} = \tilde{\mathcal{O}}\left(\frac{p' dr}{NT}\right), \tag{12}$$

where $\epsilon_{\mathrm{tr}}$ is the surplus training error of the estimate $\tilde{\boldsymbol{A}}$ over the empirical risk minimizer $\hat{\boldsymbol{A}}$.

Estimates satisfying equations (11) or (12) are computationally tractable in practice. This implies that practitioners can determine the required number of samples, or $N$ and $T$, for estimating the system parameters up to a fixed precision by using the rates in Section 4.

## 6    CONCLUSION

In this work, we extend non-asymptotic linear system identification theory and derive upper bounds on the sample complexity of learning long-context linear autoregressive models. Our bounds improve upon the existing arguments specific to first-order systems by employing a uniform concentration argument over prediction differences. We further establish improved statistical rates when learning under a low-rank assumption. Finally, we show that even with long or unbounded generative contexts, *misspecification* still allows the estimation of the matrices with a reduced sample complexity and for stable systems.

While this work makes significant progress for non-asymptotic linear system identification theory, several technical questions remain open for further investigation. Can the OLS operator norm be coarsely controlled to derive rates for unconstrained OLS in the $NT = \Omega(pdr)$ regime? Is it possible to find efficient algorithms that would benefit from low-rank assumptions? Lastly, can misspecification be beneficial for marginally stable systems?

ACKNOWLEDGMENTS

This project was supported by the Swiss National Science Foundation (grant number 212111). This work was partially funded by an unrestricted gift from Google.

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

## A  SKETCH OF PROOF

We provide a sketch of proof for Theorem 4.1. The proofs of Theorems 4.2 and 4.3 are similar and can be found in Section C. In the following, $\Delta_{\boldsymbol{A}}$ is a shorthand for $M_{\boldsymbol{A}} - M_{\boldsymbol{A}^\star}$ and $E \in \mathbb{R}^{Td \times N}$ is the matrix that collects the noise concatenated over time, as explained in Theorem C.1.

The empirical risk minimizer $\hat{\boldsymbol{A}}$ satisfies the following optimality condition:

$$\mathcal{L}(\hat{\boldsymbol{A}}) \leqslant \mathcal{L}(\boldsymbol{A}^\star), \quad \text{or written differently,} \quad \left\|\Delta_{\hat{\boldsymbol{A}}} L_\star E\right\|^2 \leqslant 2 \operatorname{Tr}\left(E^\top L_\star^\top \Delta_{\hat{\boldsymbol{A}}}^\top E\right), \quad (13)$$

due to the *well-specified* setting, i.e., $\boldsymbol{A}^\star \in \mathcal{A}(D)$. The condition in Equation (13) is of interest as

$$\forall \boldsymbol{A} \in \mathcal{A}(D), \quad \mathbb{E}\left[\left\|\Delta_{\hat{\boldsymbol{A}}} L_\star E\right\|^2\right] = \sigma^2 N \|\Delta_{\hat{\boldsymbol{A}}} L_\star\|_F^2 > 0 = \mathbb{E}\left[\operatorname{Tr}\left(E^\top L_\star^\top \Delta_{\hat{\boldsymbol{A}}}^\top E\right)\right].$$

This inequality hints that if for a set of matrices $\mathcal{A}'(D) \subseteq \mathcal{A}(D)$, there is a uniform result

$$\mathcal{E} := \left\{\forall \boldsymbol{A} \in \mathcal{A}'(D) : \left\|\Delta_{\hat{\boldsymbol{A}}} L_\star E\right\|^2 \geqslant 2 \operatorname{Tr}\left(E^\top L_\star^\top \Delta_{\hat{\boldsymbol{A}}}^\top E\right)\right\}, \quad (14)$$

with high probability as seen from their means, then the empirical risk minimizer $\hat{\boldsymbol{A}}$ belongs to the set $\mathcal{A}(D) \setminus \mathcal{A}'(D)$ with the same high probability by a simple Bayesian argument. Hence, the proof of Theorem 4.1 is reduced to proving Equation (14) for a suitable set of matrices.

Fix a $\boldsymbol{A} \in \mathcal{A}'(D)$ and study the martingale series defined through the differences sequences

$$d_{t,i}^{(n)} = \left((\boldsymbol{A} - \boldsymbol{A}^\star) X_t^{(n)}\right)_i \left(\xi_t^{(n)}\right)_i, \quad (15)$$

where the series is first ordered in $i$, then in $t$, and finally in $n$. The sum of the differences is then

$$Y_{\boldsymbol{A}} = \sum_{n,t,i} d_{t,i}^{(n)} = \sum_{n,t} \left\langle (\boldsymbol{A} - \boldsymbol{A}^\star) X_t^{(n)}, \xi_t^{(n)} \right\rangle = \operatorname{Tr}\left(E^\top L_\star^\top \Delta_{\boldsymbol{A}}^\top E\right),$$

and the quadratic predictable variation of the series is

$$W_{\boldsymbol{A}} = \sum_{n,t,i} \mathbb{E}_{\left(\xi_t^{(n)}\right)_i} \left[\left(d_{t,i}^{(n)}\right)^2\right] = \sum_{n,t} \left\|(\boldsymbol{A}^\star - \boldsymbol{A}) X_t^{(n)}\right\|^2 = \sigma^2 \|\Delta_{\boldsymbol{A}} L_\star E\|^2.$$

The condition that is asked in Equation (14) is then that the sum of the differences $Y_{\boldsymbol{A}}$ is large compared to the quadratic predictable variation $W_{\boldsymbol{A}}$, i.e., $\mathcal{E} = \left\{\forall \boldsymbol{A} \in \mathcal{A}'(D) : W_{\boldsymbol{A}} \leqslant 2\sigma^2 Y_{\boldsymbol{A}}\right\}$.

In order to prove probabilistic statements on $\mathcal{E}$, we use Freedman's inequality (Freedman, 1975; Dzhaparidze & Van Zanten, 2001) which gives control on $Y_{\boldsymbol{A}}$ and $W_{\boldsymbol{A}}^R$ for a particular $\boldsymbol{A}$:

$$\mathbb{P}\left(Y_{\boldsymbol{A}} \geqslant r_Y, W_{\boldsymbol{A}}^R \leqslant r_W\right) \leqslant \exp\left(-\frac{r_Y^2/2}{r_W + R r_Y}\right), \quad (16)$$

where $W_{\boldsymbol{A}}^R = W_{\boldsymbol{A}} + \sum_{n,t,i} \mathbb{1}_{d_{t,i}^{(n)} > R} \left(d_{t,i}^{(n)}\right)^2$ and $r_Y, r_W, R > 0$ are arbitrary constants. As the noise is sub-Gaussian, it is possible to upper bound $W_{\boldsymbol{A}}^R$ with $W_{\boldsymbol{A}}$:

$$\sum_{n,t,i} \left(d_{t,i}^{(n)}\right)^2 \leqslant \sup_{n,t,i} \left(\xi_t^{(n)}\right)_i^2 \cdot \sum_{n,t} \|(\boldsymbol{A} - \boldsymbol{A}^\star) X_t^{(n)}\|^2$$

$$\stackrel{\text{w.h.p}}{\Longrightarrow} \quad \forall \boldsymbol{A} : W_{\boldsymbol{A}}^R \leqslant C_1 \ln(2dNT) W_{\boldsymbol{A}}.$$

Further, assume that there are uniform upper and lower bounds on $Y_{\boldsymbol{A}}$ and $W_{\boldsymbol{A}}$, respectively:

$$\exists 0 < \alpha_L < \alpha_U \text{ such that } \forall \boldsymbol{A} \in \mathcal{A}'(D) : Y_{\boldsymbol{A}} \leqslant \alpha_U \text{ and } \alpha_L \leqslant W_{\boldsymbol{A}} \leqslant W_{\boldsymbol{A}}^R. \qquad (17)$$

Let $\gamma = C_1 \ln 2dNT$ and set $k'$ to be the smallest integer that verifies $e^{k'} \alpha_L \geqslant 2\gamma\sigma^2 \alpha_U$. Then,

$$\mathbb{P}\left(W_{\boldsymbol{A}} \leqslant 2\sigma^2 Y_{\boldsymbol{A}}\right) \leqslant \mathbb{P}\left(W_{\boldsymbol{A}}^R \leqslant 2\gamma\sigma^2 Y_{\boldsymbol{A}}\right) \leqslant \cup_{k=1}^{k'} \mathbb{P}\left(W_{\boldsymbol{A}}^R \leqslant e^k \alpha_L, 2\gamma\sigma^2 Y_{\boldsymbol{A}} \geqslant e^{k-1}\alpha_L\right).$$

Each of the terms in the union can be controlled by the Freedman's inequality in Equation (16) with the choices of $r_Y = \alpha_L e^{k-1}/2\gamma\sigma^2$, $r_W = \alpha_L e^k$ and $R = 2e\gamma\sigma^2$:

$$
\begin{aligned}
\mathbb{P}\left(W_{\boldsymbol{A}} \leqslant 2\gamma\sigma^2 Y_{\boldsymbol{A}}\right) &\leqslant \sum_{k=1}^{k'} \exp\left(-\frac{e^{k-2}\alpha_L}{8e\gamma^2\sigma^4}\right) \\
&\leqslant \exp\left(-\frac{\alpha_L}{C_2\sigma^4 (\ln 2dNT)^2} + \ln\left(\ln\left(\frac{C_3\sigma^2 \ln(2dNT)\alpha_U}{\alpha_L}\right) + 1\right)\right).
\end{aligned}
\qquad (18)
$$

As can be seen from Equation (18), the probability of the event $\{W_{\boldsymbol{A}} \leqslant 2\sigma^2 Y_{\boldsymbol{A}}\}$ is largely controlled with the lower bound $\alpha_L$ as the ratio $\alpha_U/\alpha_L$ only matters logarithmically. This is crucial as the $\alpha_U$ and $\alpha_L$ differ with the condition number $\kappa$, which can scale with $T$.

Therefore, it possible to control the event $\mathcal{E}$ with a union bound over an $\epsilon$-net of $\mathcal{A}'(D)$. In particular, $\alpha_L$ needs to be uniformly bounded below as follows:

$$\alpha_L \geqslant \ln|\mathcal{N}_\epsilon(\mathcal{A}'(D))| \ln(2dNT)^2.$$

This is achieved by Hanson-Wright inequality (Hanson & Wright, 1971) which allows us to derive the needed uniform lower and upper bounds in Equation (17)

$$Y_{\boldsymbol{A}} \leqslant c_1\sigma^2 \|L_\star\|_{\mathrm{op}} \sqrt{pdrN} \|\Delta_{\boldsymbol{A}}\|_F, \quad W_{\boldsymbol{A}}^R \geqslant W_{\boldsymbol{A}} \geqslant c_2\sigma^4 \sigma_{\min}(L_\star)^2 N \|\Delta_{\boldsymbol{A}}\|_F^2,$$

with high probability as long as $\mathcal{A}'(D)$ is composed of matrices that satisfy

$$\frac{\|\Delta_{\boldsymbol{A}}\|_F^2}{\|\Delta_{\boldsymbol{A}}\|_{\mathrm{op}}^2} \geqslant \ln|\mathcal{N}_\epsilon(\mathcal{A}'(D))|, \quad \text{where} \quad \epsilon \sim \frac{\text{polysqrt}(p, d, N, T, \kappa)}{1 + c^2 \ln\frac{1}{\delta}}.$$

Here, we pick up the dependency on $\kappa$ as $\epsilon$ scales with $\kappa$. This is needed to bound the worst-case errors while transitioning from point-wise bounds on the $\epsilon$-net $\mathcal{N}_\epsilon(A'(D))$ to the whole set $A'(D)$.

Finally, since there is a uniform bound on $\|\Delta_{\boldsymbol{A}}\|_{\mathrm{op}} \leqslant 2D$ implied by Theorem 3.4, setting

$$\mathcal{A}'(D) = \left\{\boldsymbol{A} \mid \|\Delta_{\boldsymbol{A}}\|_F^2 \geqslant C(\delta)D^2 \frac{pdr}{N} \text{polylog}(\kappa, p, d, N, T)\right\},$$

for some constant $C(\delta)$ is sufficient to deduce $\hat{\boldsymbol{A}} \in \mathcal{A}(D) \setminus \mathcal{A}'(D)$ with probability $1-\delta$. The proof of Theorem 4.1 is then complete as $\|\Delta_{\boldsymbol{A}}\|_F^2 = \|M_{\boldsymbol{A}} - M_{\boldsymbol{A}^\star}\|_F^2 \geqslant (T-p)\|\boldsymbol{A} - \boldsymbol{A}^\star\|_F^2$.

# B  PRELIMINARY TOOLS

## B.1  HANSON-WRIGHT INEQUALITY

We use Hanson-Wright inequality (Hanson & Wright, 1971; Wright, 1973; Rudelson & Vershynin, 2013) to show concentration of certain second-order terms.

**Theorem B.1.** *(Hanson-Wright) Let $Z = (Z_1, \ldots, Z_n) \in \mathbb{R}^n$ be a random vector with independent components $Z_i$ which satisfy $\mathbb{E}[Z_i] = 0$ and $\|Z_i\|_{\psi_2} \leqslant K$. Let $P$ be an $n \times n$ matrix. Then, for every $r \geqslant 0$,*

$$\mathbb{P}\left(\left|Z^\top P Z - \mathbb{E}\left[Z^\top P Z\right]\right| > r\right) \leqslant 2\exp\left(-C_{HW}\min\left(\frac{r^2}{K^4\|P\|_F^2}, \frac{r}{K^2\|P\|_{\mathrm{op}}}\right)\right).$$

*The bound can be turned into a one-sided bound by dropping the constant 2.*

**Remark B.2.** *For data regime considered in this paper, $K = c\sigma$ in Theorem B.1.*

## B.2 FREEDMAN'S INEQUALITY

We use an extension of Freedman's inequality (Freedman, 1975) to non-bounded differences by Dzhaparidze & Van Zanten (2001) to show concentration of certain second-order terms. For the sake of completeness, we provide the original Freedman's inequality. We also remark that it is possible to use the original Freedman's inequality in our proofs to deal with *any bounded noise*, leading to improved logarithmic factors, as explained in Theorem C.27.

**Theorem B.3.** *(Freedman's inequality) Let $Y_0, \ldots, Y_n$ be a real-valued martingale series that is adapted to the filtration $\mathcal{F}_0, \ldots, \mathcal{F}_n$ where $Y_0 = 0$. Let $d_1, \ldots, d_n$ be the difference sequence induced, i.e.,*

$$d_i = Y_i - Y_{i-1} \quad for \ i = 1, \ldots, n.$$

*Assume that $d_i$ is upper bounded by some $R$, i.e., $|d_i| \leqslant R$ for all $i$. Let $W_i$ be the quadratic variation of the martingale series, i.e.,*

$$W_i = \sum_{j=1}^{i} \mathbb{E}[d_j^2 \mid \mathcal{F}_{j-1}] \quad for \ i = 1, \ldots, n.$$

*Then, for any $r, W > 0$,*

$$\mathbb{P}\left(\exists k \geqslant 0 : Y_k \geqslant r \ and \ W_k \leqslant W\right) \leqslant \exp\left(-\frac{r^2/2}{W + Rr}\right).$$

**Theorem B.4.** *(Freedman's inequality for non-bounded differences) Let $Y_0, \ldots, Y_n$ be a real-valued martingale series that is adapted to the filtration $\mathcal{F}_0, \ldots, \mathcal{F}_n$ where $Y_0 = 0$. Let $d_1, \ldots, d_n$ be the difference sequence induced, i.e.,*

$$d_i = Y_i - Y_{i-1} \quad for \ i = 1, \ldots, n.$$

*Let $W_i^R$ be the quadratic variation of the martingale series plus an error term for large differences,*

$$W_i^R = \sum_{j=1}^{i} \mathbb{E}[d_j^2 \mid \mathcal{F}_{j-1}] + d_i^2 \mathbb{1}_{\{|d_i|>R\}} \quad for \ i = 1, \ldots, n.$$

*We set $W_i = W_i^0$ for ease of notation. Then, for any $r, R, W > 0$,*

$$\mathbb{P}\left(\exists k \geqslant 0 : Y_k \geqslant r \ and \ W_k^R \leqslant W\right) \leqslant \exp\left(-\frac{r^2/2}{W + Rr}\right).$$

We extend these theorems in Theorem B.5 to compare the quadratic variation with the martingale series itself. This is useful in our proofs to show certain events necessarily implied by empirical risk minimization do not occur with high probability.

**Lemma B.5.** *Let $\gamma, R > 0, \alpha_U \geqslant \alpha_L > 0$ be scalars and let $\mathcal{E}$ denote the following event*

$$\mathcal{E} = \left\{W_n^R \geqslant \alpha_L\right\} \cap \left\{Y_n \leqslant \alpha_U\right\},$$

*where $Y_n$ and $W_n^R$ verifies the assumptions of Theorem B.4. Then, we have the following concentration inequality*

$$\mathbb{P}\left(\left\{W_n^R \leqslant \gamma Y_n\right\} \cap \mathcal{E}\right) \leqslant \exp\left(-\frac{\alpha_L}{2e\gamma\left(e\gamma + R\right)} + \ln\left(\ln\left(\frac{\gamma\alpha_U}{\alpha_L}\right) + 1\right)\right).$$

*Proof.* Let $\mathcal{G} = \left\{\alpha_L, e\alpha_L, \ldots, e^k \alpha_L\right\}$ where $k$ is the smallest positive integer such that

$$e^k \alpha_L \geqslant \gamma\alpha_U.$$

Then, by a union bound,

$$\begin{aligned}
\mathbb{P}\left(W_n^R \leqslant \gamma Y_n \cap \mathcal{E}\right) &\leqslant \mathbb{P}\left(\cup_{i=1}^{k}\left(\left\{W_n^R \leqslant e^i \alpha_L, \gamma Y_n \geqslant e^{i-1}\alpha_L\right\} \cap \mathcal{E}\right)\right) \\
&\leqslant \mathbb{P}\left(\cup_{i=1}^{k}\left(\left\{W_n^R \leqslant e^i \alpha_L, \gamma Y_n \geqslant e^{i-1}\alpha_L\right\}\right)\right) \\
&\leqslant \sum_{i=1}^{k} \mathbb{P}\left(\left\{W_n^R \leqslant e^i \alpha_L, \gamma Y_n \geqslant e^{i-1}\alpha_L\right\}\right).
\end{aligned}$$

By applying Theorem B.4 with $r = e^{i-1}\alpha_L/\gamma$ and $W = e^i\alpha_L$, we obtain

$$\mathbb{P}\left(W_n^R \leqslant e^i\alpha_L, Y_n \geqslant e^{i-1}\alpha_L/\gamma\right) \leqslant \exp\left(-\frac{e^{i-2}\alpha_L}{2\gamma(e\gamma+R)}\right),$$

for each $i = 1, \ldots, k$. The union bound gives

$$\sum_{i=1}^k \mathbb{P}\left(\{W_n^R \leqslant e^i\alpha_L, Y_n \geqslant e^{i-1}\alpha_L\}\right) \leqslant \sum_{i=1}^k \exp\left(-\frac{e^{i-2}\alpha_L}{2\gamma(e\gamma+R)}\right)$$

$$\leqslant \exp\left(-\frac{\alpha_L}{2e\gamma(e\gamma+R)} + \ln k\right).$$

The result follows by noting that

$$k \leqslant \ln\left(\frac{\gamma\alpha_U}{\alpha_L}\right) + 1.$$

$\square$

## B.3 SUPREMUM OF THE NOISE

We need the following lemma to control the supremum of the noise in our proofs.

**Lemma B.6.** *Let $X_1, \ldots, X_n$ be i.i.d. mean zero and $\sigma^2$-sub-Gaussian random variables (in the sense provided in Theorem 3.1). Then, there exists a universal constant $c'$ such that for any $t > 0$,*

$$\mathbb{P}\left(\sup_{i=1,\ldots,n}|X_i| > c'\sigma\sqrt{2\ln 2n} + t\right) \leqslant \exp\left(-\frac{t^2}{2c'\sigma^2}\right).$$

*Proof.* By the sub-Gaussian property, we have a universal constant $c'$ such that

$$\mathbb{P}(X_i \geqslant r) \leqslant \exp\left(-\frac{r^2}{2c'^2\sigma^2}\right).$$

Then, by the union bound,

$$\mathbb{P}\left(\sup_{i=1,\ldots,n} X_i \geqslant r\right) \leqslant \cup_i \mathbb{P}(X_i \geqslant r) \leqslant n\exp\left(-\frac{r^2}{2c'^2\sigma^2}\right).$$

Similarly, we have

$$\mathbb{P}\left(\sup_{i=1,\ldots,n} -X_i \geqslant r\right) \leqslant \cup_i \mathbb{P}(-X_i \geqslant r) \leqslant n\exp\left(-\frac{r^2}{2c'^2\sigma^2}\right).$$

The result follows by union bound with $r = c'\sigma\sqrt{2\ln 2n} + t$. $\square$

**Corollary B.7.** *For any $0 < \delta < e^{-1}$, there exists a universal constant $c'$ such that*

$$\sup_{t,n}\|\xi_t^{(n)}\|_\infty \leqslant c'\sigma\sqrt{2}\left(\sqrt{\ln 2dTN} + \sqrt{\ln\frac{1}{\delta}}\right),$$

*with probability $1 - \delta$.*

*Proof.* Each component $\left(\xi_t^{(n)}\right)_i$ are i.i.d. and sub-Gaussian with parameter $\sigma$. By Theorem B.6,

$$\mathbb{P}\left(\sup_{t,n}\|\xi_t^{(n)}\|_\infty > c'\sigma\sqrt{2\ln 2dTN} + t\right) \leqslant \exp\left(-\frac{t^2}{2c'^2\sigma^2}\right),$$

for any $t > 0$. Select $t = c'\sigma\sqrt{2\ln\frac{1}{\delta}}$ to obtain the desired confidence level of $\delta$. $\square$

## B.4 COVERING NUMBERS

We use the following lemma by (Candès & Plan, 2011) to control the covering numbers of the set of matrices:

**Lemma B.8.** *(Covering number for low-rank matrices) Let* $\mathcal{M}_r^{n_1 \times n_2}$ *denote the set of low-rank matrices of rank* $r$ *and Frobenius norm bounded by* $1$:

$$\mathcal{M}_r^{n_1 \times n_2} = \left\{ M \in \mathbb{R}^{n_1 \times n_2} : \|M\|_F = 1, \operatorname{rank}(M) \leqslant r \right\}.$$

*Then, there exists an* $\epsilon$-net $\mathcal{S}$ *of* $\mathcal{M}$ *for Frobenius norm such that*

$$|\mathcal{S}| \leqslant \left( \frac{9}{\epsilon} \right)^{(n_1 + n_2 + 1)r}.$$

**Corollary B.9.** *Let* $\mathcal{M}_{r,p'}(D)$ *denote the following set of matrices with bounded operator norm:*

$$\mathcal{M}_{r,p'}(D) = \left\{ \boldsymbol{A} \in \mathbb{R}^{d \times pd} \mid \|\boldsymbol{A}\|_{\mathrm{op}} \leqslant D, \ \operatorname{rank}(A_i) \leqslant r, \ A_{p'+1} = \cdots = A_p = 0 \right\}.$$

*Then, there exists an* $\epsilon$-net $\mathcal{S}$ *of* $\mathcal{M}_{r,p'}(D)$ *for operator norm such that*

$$|\mathcal{S}| \leqslant \exp \left( 9p'dr \ln \frac{Dp'}{\epsilon} \right).$$

*Proof.* Since for any matrix $M$, $\|M\|_{\mathrm{op}} \leqslant \|M\|_F$, it is sufficient to give a covering number for the Frobenius norm. Let $\mathcal{M}_r^{d \times d}(D)$ be the set of low-rank matrices of rank $r$ and Frobenius norm bounded by $D$:

$$\mathcal{M}_r^{d \times d}(D) = \left\{ M \in \mathbb{R}^{d \times d} \mid \|M\|_F \leqslant D, \ \operatorname{rank}(M_i) \leqslant r \right\}.$$

By Theorem B.8, there exists an $\frac{\epsilon}{p'}$-net $\mathcal{S}$ of $\mathcal{M}_r^{d \times d}(D)$ for Frobenius norm such that

$$|\mathcal{S}| \leqslant \left( \frac{9Dp'}{\epsilon} \right)^{(2d+1)r} \leqslant \exp \left( 9dr \ln \frac{Dp'}{\epsilon} \right),$$

as any $\frac{\epsilon}{Dp'}$-net of $\mathcal{M}_r^{d \times d}(1)$ gives an $\frac{\epsilon}{p'}$-net of $\mathcal{M}_r^{d \times d}(D)$.

Observe that the set $\mathcal{M}_{r,p'}(D)$ is a subset of

$$\mathcal{U} = \left( \mathcal{M}_r^{d \times d}(D) \right)^{p'} \times \{0_{d \times d}\}^{p - p'}.$$

Then, $(\mathcal{S})^{p'}$ gives an $\epsilon$-net of $\mathcal{U}$ and hence of $\mathcal{M}_{r,p'}(D)$. □

## B.5 PROOF OF THEOREM 3.5

*Proof.* Using Theorem C.12, we have $\|M_{\boldsymbol{A}^\star}\|_{\mathrm{op}} \leqslant \sqrt{p} \|\boldsymbol{A}^\star\|_{\mathrm{op}}$, directly leading to **(ii)**.

For **(i)**, we have

$$\|M_{\boldsymbol{A}^\star}\|_{\mathrm{op}} \leqslant \sum_{i=1}^p \|M_{\boldsymbol{A}^{\star,(i)}}\|_{\mathrm{op}}, \quad \text{where} \quad \boldsymbol{A}^{\star,(i)} = \left( \underbrace{0_d, 0_d, \ldots, 0_d}_{i-1 \text{ times}}, \boldsymbol{A}^{\star,i}, 0_d, \ldots, 0_d \right).$$

Then, it is easy to see that

$$\|M_{\boldsymbol{A}^{\star,(i)}}\|_{\mathrm{op}} \leqslant \|\boldsymbol{A}_i^\star\|_{\mathrm{op}}.$$

□

## C   PROOF OF THEOREMS 4.1 TO 4.3

Before proving the main theorems, we recall certain definitions from the main body of the paper:

**Definition C.1.** *For any $\boldsymbol{A} \in \mathbb{R}^{d \times pd}$, let $\Delta_{\boldsymbol{A}} = \Delta_{\boldsymbol{A},p'}$ be defined as follows*

$$\Delta_{\boldsymbol{A},i} = (M_{\boldsymbol{A}_i} - M_{\boldsymbol{A}_i^\star}),$$

$$\text{where} \quad \boldsymbol{A}_i = (A_1, \ldots, A_i, 0_d, \ldots, 0_d), \; \boldsymbol{A}_i^\star = (A_1^\star, \ldots, A_i^\star, 0_d, \ldots, 0_d).$$

*Let $\xi^{(i)} \in \mathbb{R}^{Td}$ be the whole noise concatenated in time, i.e.,*

$$\xi^{(i)} = \left( \xi_1^{(i)}, \ldots, \xi_T^{(i)} \right),$$

*and let $E \in \mathbb{R}^{Td \times N}$ be the matrix that collects the noise for all sequences, i.e.,*

$$E = \left( \xi^{(1)}, \ldots, \xi^{(n)} \right).$$

**Proposition C.2.** *With the definitions of Theorem C.1, we have the following properties:*

$$\sum_{n,t} \langle (\boldsymbol{A}_i - \boldsymbol{A}_i^\star) X_t^{(n)}, \xi_t^{(n)} \rangle = \text{Tr}(E^\top \Delta_{\boldsymbol{A},i} L_\star E),$$

$$\sum_{n,t} \left\| (\boldsymbol{A}_i - \boldsymbol{A}_i^\star) X_t^{(n)} \right\|^2 = \left\| (M_{\boldsymbol{A}_i} - M_{\boldsymbol{A}_i^\star}) L_\star E \right\|_F^2 = \left\| \Delta_{\boldsymbol{A},i} L_\star E \right\|_F^2.$$

**Definition C.3.** *Let $\mathcal{A}_{r,p}(D)$ and $\mathcal{S}_{r,p}(C,D)$ be the search and solution set for constants $C, D \geqslant 1$:*

$$\mathcal{A}_{r,p'}(D) = \left\{ \boldsymbol{A} \in \mathbb{R}^{d \times pd} \; \middle| \; \|\Delta_{\boldsymbol{A}}\|_{\text{op}} \leqslant D, \; \text{rank}(A_i) \leqslant r, \; A_{p'+1} = \cdots = A_p = 0 \right\},$$

$$\mathcal{S}_{r,p'}(C,D) = \left\{ \boldsymbol{A} \in \mathcal{A}(D) \; \middle| \; \|\boldsymbol{A} - \boldsymbol{A}_{p'}^\star\|_F^2 \leqslant CD^2 \eta^2 \tau \frac{p' dr}{N(T - p')} \right\},$$

*where $\eta$ is a constant that captures an additional factor for the misspecified setting,*

$$\eta = \begin{cases} 1 & \text{if } p' = p, \\ \max \left\{ 1, 1 + \left\| \left( M_{\boldsymbol{A}^\star} - M_{\boldsymbol{A}_{p'}^\star} \right) L_\star \right\|_{\text{op}} \right\} & \text{if } p' < p, \end{cases}$$

*and $\tau$ is the following logarithmic term:*

$$\tau = \left( 1 + \ln \frac{p'^2 dNT^2}{T - p'} \right)^3 (1 + \ln \kappa).$$

*Let $\mathcal{G}_{r,p'}(C,D)$ be defined as follows,*

$$\mathcal{G}_{r,p'}(C,D) = \left\{ \boldsymbol{A} \in \mathcal{A}_{r,p}(D) \; \middle| \; \frac{\|\Delta_{\boldsymbol{A}}\|_F^2}{\|\Delta_{\boldsymbol{A}}\|_{\text{op}}^2} \leqslant C\eta^2 \tau \frac{p' dr}{N} \right\}.$$

*We set $\mathcal{A}_{r,p'} = \mathcal{A}_{r,p'}(\infty)$ and $\mathcal{G}(C)_{r,p'} = \mathcal{G}(C,\infty)$. Lastly, we drop the subscript $r, p'$ when it is clear from the context.*

### C.1   MAIN RESULTS

In this subsection, we state Theorem C.4 that generalizes the statements in Theorems 4.1 to 4.3. We give a proof that reduces Theorem C.4 to a uniform concentration result in Theorem C.5. The proof of Theorem C.5 is deferred to Section C.5. Next, in Theorem C.6, we discuss the factor $\eta$ that appears in our results and the conditions under which our misspecification bounds are tight. Lastly, Theorem C.7 removes the constraint on the diameter of the search set which allows us to treat OLS as a special case of the main theorem.

**Theorem C.4.** *Let Theorems 3.1 and 3.4 hold. Furthermore, let Theorem 3.7 for $r < d$ and Theorem 3.9 for $p' < p$ hold. Consider the following constrained empirical risk minimizer:*

$$\hat{\boldsymbol{A}} = \text{argmin}_{\boldsymbol{A} \in \mathcal{A}(D)} \mathcal{L}(\boldsymbol{A}).$$

*Then, for any $0 < \delta < e^{-1}$, there exists a constant $C(\delta)$ such that*

$$\mathbb{P} \left( \hat{\boldsymbol{A}} \in \mathcal{S}(C(\delta), D) \right) \geqslant 1 - \delta.$$

*Proof.* Let $\mathcal{E}_{\boldsymbol{A}}$ be the following event

$$\mathcal{E}_{\boldsymbol{A}} = \{\|\Delta_{\boldsymbol{A}} L_\star E\|_F^2 \leqslant 2\eta \operatorname{Tr}(E^\top \Delta_{\boldsymbol{A}} L_\star E)\}.$$

By Theorem C.13, $\mathcal{G}(C, D) \subset \mathcal{S}(C, D)$ and thus, for any random choice of $\boldsymbol{A}$,

$$\mathbb{P}\left(\{\boldsymbol{A} \in \mathcal{S}(C, D)\}\right) \geqslant \mathbb{P}\left(\{\boldsymbol{A} \in \mathcal{G}(C, D)\}\right) = 1 - \mathbb{P}\left(\{\boldsymbol{A} \in \mathcal{A}(D) \setminus \mathcal{G}(C, D)\}\right).$$

For the choice of $\hat{\boldsymbol{A}}$, $\mathbb{P}\left(\mathcal{E}_{\hat{\boldsymbol{A}}}\right) = 1$ by Theorem C.15 and

$$\mathbb{P}\left(\left\{\hat{\boldsymbol{A}} \in \mathcal{S}(C, D)\right\}\right) \geqslant 1 - \mathbb{P}\left(\left\{\hat{\boldsymbol{A}} \in \mathcal{A}(D) \setminus \mathcal{G}(C, D)\right\} \mid \mathcal{E}_{\hat{\boldsymbol{A}}}\right).$$

By Bayes rule, we have

$$\mathbb{P}\left(\left\{\hat{\boldsymbol{A}} \in \mathcal{S}(C, D)\right\}\right) \geqslant 1 - \mathbb{P}\left(\mathcal{E}_{\hat{\boldsymbol{A}}} \mid \left\{\hat{\boldsymbol{A}} \in \mathcal{A}(D) \setminus \mathcal{G}(C, D)\right\}\right) \mathbb{P}\left(\left\{\hat{\boldsymbol{A}} \in \mathcal{A}(D) \setminus \mathcal{G}(C, D)\right\}\right)$$

$$\geqslant 1 - \mathbb{P}\left(\mathcal{E}_{\hat{\boldsymbol{A}}} \mid \left\{\hat{\boldsymbol{A}} \in \mathcal{A}(D) \setminus \mathcal{G}(C, D)\right\}\right).$$

Then, the proof is complete by applying Theorem C.5 to the right-hand side. $\square$

**Theorem C.5.** *Let all the assumptions of Theorem C.4 hold. Then, for any $0 < \delta < e^{-1}$, there exists a constant $C(\delta)$ such that*

$$\mathbb{P}\left(\exists \boldsymbol{A} \in \mathcal{A}(D) \setminus \mathcal{G}(C(\delta), D) : \|\Delta_{\boldsymbol{A}} L_\star E\|_F^2 \leqslant 2\eta \operatorname{Tr}(E^\top \Delta_{\boldsymbol{A}} L_\star E)\right) \leqslant \delta. \tag{19}$$

**Remark C.6.** *For strictly stable systems with $\|M_{\boldsymbol{A}^\star}\|_{\mathrm{op}} < 1$ and $\|M_{\boldsymbol{A}_{p'}^\star}\|_{\mathrm{op}} < 1$, the factor $\eta$ is controlled by Theorem C.10. However, for marginally stable systems or explosive systems, there is no a prior good upper bound on $\eta$, implying that the misspecification results only applies to strictly stable systems without any further assumptions.*

**Corollary C.7.** *Let all the assumptions of Theorem C.4 hold and suppose that $NT$ verifies the following condition:*

$$N(T - p') \geqslant C(\delta)\eta^2 \tau p'^2 dr, \tag{20}$$

*where $0 < \delta < e^{-1}$ is a constant and $C(\delta)$ is given by Theorem C.4 Then, the OLS estimator*

$$\hat{\boldsymbol{A}}_{\mathrm{OLS}} = \operatorname{argmin}_{\boldsymbol{A} \in \mathcal{A}(\infty)} \mathcal{L}(\boldsymbol{A}),$$

*satisfies the same concentration result as in Theorem C.4:*

$$\mathbb{P}\left(\hat{\boldsymbol{A}}_{\mathrm{OLS}} \in \mathcal{S}(C(\delta), D)\right) \geqslant 1 - \delta.$$

*Proof.* Assume that $D$ is sufficiently large such that

$$\mathcal{A}(D)^{\mathrm{o}} \subset \mathcal{S}(C(\delta), D),$$

i.e., the interior of $\mathcal{A}(D)$ contains $\mathcal{S}(C(\delta), D)$. We have the following relation:

$$\mathcal{A}(\infty) = \left\{\boldsymbol{A}' = \alpha \boldsymbol{A} \mid \boldsymbol{A} \in \mathcal{A}(D) \setminus \mathcal{S}(C, D), \alpha \geqslant 1 \in \mathbb{R}\right\}.$$

Then, by Theorem C.5, we have

$$\mathbb{P}\left(\exists \boldsymbol{A} \in \mathcal{A}(\infty) \setminus \mathcal{S}(C(\delta), D) : \|\Delta_{\boldsymbol{A}} L_\star E\|_F^2 \leqslant 2\eta \operatorname{Tr}(E^\top \Delta_{\boldsymbol{A}} L_\star E)\right)$$

$$= \mathbb{P}\left(\exists \alpha \geqslant 1 \in \mathbb{R}, \boldsymbol{A} \in \mathcal{A}(D) \setminus \mathcal{S}(C(\delta), D) : \|\Delta_{\alpha \boldsymbol{A}} L_\star E\|_F^2 \leqslant 2\eta \operatorname{Tr}(E^\top \Delta_{\alpha \boldsymbol{A}} L_\star E)\right)$$

$$= \mathbb{P}\left(\exists \boldsymbol{A} \in \mathcal{A}(D) \setminus \mathcal{S}(C(\delta), D) : \|\Delta_{\boldsymbol{A}} L_\star E\|_F^2 \leqslant 2\eta \operatorname{Tr}(E^\top \Delta_{\boldsymbol{A}} L_\star E)\right) \leqslant \delta,$$

as $\forall \alpha \geqslant 1$, we have the following:

$$\|\Delta_{\boldsymbol{A}} L_\star E\|_F^2 \leqslant 2\eta \operatorname{Tr}(E^\top \Delta_{\boldsymbol{A}} L_\star E) \implies \|\Delta_{\alpha \boldsymbol{A}} L_\star E\|_F^2 \leqslant 2\eta \operatorname{Tr}(E^\top \Delta_{\alpha \boldsymbol{A}} L_\star E).$$

Thus, the result is complete by applying the same argument as in Theorem C.4 where $\mathcal{A}(D)$ is replaced by $\mathcal{A}(\infty)$.

We only need to provide a $D$ such that $\mathcal{A}(D)^{\mathrm{o}} \subset \mathcal{S}(C(\delta), D)$. For any $\boldsymbol{A} \in \mathcal{S}(C(\delta), D)$, we have

$$\|\Delta_{\boldsymbol{A}}\|_{\mathrm{op}}^2 \leqslant p' \|\boldsymbol{A}\|_{\mathrm{op}}^2 \leqslant p' \|\boldsymbol{A}\|_F^2 \leqslant C(\delta) D^2 \eta^2 \tau \frac{p'^2 dr}{N(T - p')},$$

from Theorem C.12. Therefore, we need to find a $D$ such that

$$D^2 \geqslant C(\delta) D^2 \eta^2 \tau \frac{p'^2 dr}{N(T - p')},$$

which equivalent to the condition in Equation (20). $\square$

## C.2 TECHNICAL LEMMAS

In this subsection, we present simple technical results on $L_\star, M_A$ and $\Delta_A$ that are used in the proof of Theorem C.4.

**Lemma C.8.** $L_\star$ and $M_{A^\star}$ satisfy the following relations:

$$L_\star = M_{A^\star} L_\star + I, \quad M_{A^\star} = (L_\star - I)L_\star^{-1}, \quad L_\star = (I - M_{A^\star})^{-1}.$$

*Proof.* The first relation follows from a direct computation. For the second, note that $L_\star$ is invertible since it is a lower triangular matrix with non-zero diagonals. Lastly,

$$
\begin{aligned}
L_\star &= I + M_{A^\star} L_\star \\
&= I + M_{A^\star} + M_{A^\star}^2 L_\star \\
&= \cdots \\
&= I + M_{A^\star} + M_{A^\star}^2 + \cdots + M_{A^\star}^{T-1} \\
&= (I - M_{A^\star})^{-1},
\end{aligned}
$$

where we have used the fact that $M_{A^\star}^T = 0_{Td}$. □

**Lemma C.9.** *Assume that* $\|M_{A^\star}\|_{\mathrm{op}} < 1$. *Then, the operator norm and minimum singular value of* $L_\star$ *are bounded as follows,*

$$\frac{1}{1 + \|M_{A^\star}\|_{\mathrm{op}}} \leqslant \|L_\star\|_{\mathrm{op}} \leqslant \frac{1}{1 - \|M_{A^\star}\|_{\mathrm{op}}}, \qquad \frac{1}{2} \leqslant \sigma_{\min}(L_\star).$$

*Proof.* By Weyl's inequality for singular values on the identity $L_\star = M_{A^\star} L_\star + I$ from Theorem C.8,

$$
\begin{aligned}
\|L_\star\|_{\mathrm{op}} &\leqslant \|I\|_{\mathrm{op}} + \|M_{A^\star} L_\star\|_{\mathrm{op}} \leqslant 1 + \|M_{A^\star}\|_{\mathrm{op}} \|L_\star\|_{\mathrm{op}}, \\
\|L_\star\|_{\mathrm{op}} &\geqslant \|I\|_{\mathrm{op}} - \|M_{A^\star} L_\star\|_{\mathrm{op}} \geqslant 1 - \|M_{A^\star}\|_{\mathrm{op}} \|L_\star\|_{\mathrm{op}}.
\end{aligned}
$$

This implies the desired inequalities for $\|L_\star\|_{\mathrm{op}}$. For the lower bound on minimal singular value, use Theorem C.8,

$$\sigma_{\min}(L_\star) = \sigma_{\min}\left((I - M_{A^\star})^{-1}\right) = \frac{1}{\|I - M_{A^\star}\|_{\mathrm{op}}} \geqslant \frac{1}{1 + \|M_{A^\star}\|_{\mathrm{op}}} \geqslant \frac{1}{2}.$$

□

**Corollary C.10.** *Assume that* $\|M_{A^\star}\|_{\mathrm{op}} < 1$ *and* $\|M_{A^\star_{p'}}\|_{\mathrm{op}} < 1$. *Then, we have*

$$\eta \leqslant \frac{2}{1 - \|M_{A^\star}\|_{\mathrm{op}}}.$$

*Proof.* Applying Theorem C.9,

$$\eta \leqslant \left\|M_{A^\star} - M_{A^\star_{p'}}\right\|_{\mathrm{op}} \|L_\star\|_{\mathrm{op}} \leqslant \frac{\|M_{A^\star}\|_{\mathrm{op}} + \left\|M_{A^\star_{p'}}\right\|_{\mathrm{op}}}{1 - \|M_{A^\star}\|_{\mathrm{op}}} \leqslant \frac{2}{1 - \|M_{A^\star}\|_{\mathrm{op}}}.$$

□

**Lemma C.11.** *Assume that* $\|M_{A^\star}\|_{\mathrm{op}} \leqslant D$. *Then, the operator norm and minimum singular value of* $L_\star$ *are bounded as follows,*

$$\|L_\star\|_{\mathrm{op}} \leqslant \frac{D^T - 1}{D - 1}, \qquad \frac{1}{D + 1} \leqslant \sigma_{\min}(L_\star).$$

*Proof.* By Weyl's inequality for singular values on the identity $L_\star = I + M_{A^\star} + \cdot + M_{A^\star}^{T-1}$ from Theorem C.8,

$$\|L_\star\|_{\mathrm{op}} \leqslant \|I\|_{\mathrm{op}} + \sum_{t=1}^{T-1} \|M_{A^\star}^t\|_{\mathrm{op}} \leqslant \sum_{t=0}^{T-1} D^t \leqslant \frac{D^T - 1}{D - 1}.$$

For the lower bound on minimal singular value, use Theorem C.8,

$$\sigma_{\min}\left(L_{\star}\right) = \sigma_{\min}\left((I - M_{\boldsymbol{A}^{\star}})^{-1}\right) = \frac{1}{\|I - M_{\boldsymbol{A}^{\star}}\|_{\mathrm{op}}} \geqslant \frac{1}{1 + \|M_{\boldsymbol{A}^{\star}}\|_{\mathrm{op}}} \geqslant \frac{1}{D+1} \,.$$

$\square$

**Lemma C.12.** *For any $\boldsymbol{A} \in \mathbb{R}^{d \times pd}$,*

$$\|\boldsymbol{A}\|_{\mathrm{op}} \leqslant \|M_{\boldsymbol{A}}\|_{\mathrm{op}} \leqslant \sqrt{p'}\|\boldsymbol{A}\|_{\mathrm{op}} \,,$$

$$\frac{1}{T}\|\Delta_{\boldsymbol{A}}\|_F^2 \leqslant \|\boldsymbol{A} - \boldsymbol{A}_{p'}^{\star}\|_F^2 \leqslant \frac{1}{T - p'}\|\Delta_{\boldsymbol{A}}\|_F^2 \,.$$

*Proof.* Let $u = (u_1, \ldots, u_T) \in \mathbb{R}^{Td}$ be an arbitrary vector with $\|u\|_2^2 = 1$. Then, setting $u_{-a} = 0$ for any $a \geqslant 0$,

$$\begin{aligned}
\|M_{\boldsymbol{A}}u\|_2^2 &= \sum_{i=1}^{T} \|\left(M_{\boldsymbol{A}}u\right)_i\|_2^2 = \sum_{i=1}^{T} \|\sum_{k=1}^{p} A_k u_{i-k}\|_2^2 \\
&= \sum_{i=1}^{T} \|\boldsymbol{A}_{:p'} u_{i-p':i-1}\|_2^2 \leqslant \sum_{i=1}^{T} \|\boldsymbol{A}_{:p'}\|_{\mathrm{op}}^2 \|u_{i-p':i-1}\|_2^2 \\
&\leqslant \|\boldsymbol{A}\|_{\mathrm{op}}^2 \sum_{i=1}^{T} p' \|u_i\|_2^2 = p' \|\boldsymbol{A}\|_{\mathrm{op}}^2.
\end{aligned}$$

The left-hand side of the first inequality follows by picking $u_{p'+1:T} = 0$ and $u_{1:p'}$ as the maximal singular vector of $\boldsymbol{A}_{:p'}$ with unit length. The second inequality follows by a simple computation. $\square$

**Corollary C.13.** *For any $\boldsymbol{A} \in \mathcal{A}(D)$,*

$$\|\boldsymbol{A} - \boldsymbol{A}_{p'}^{\star}\|_F^2 \leqslant \frac{D^2}{T - p'} \frac{\|\Delta_{\boldsymbol{A}}\|_F^2}{\|\Delta_{\boldsymbol{A}}\|_{\mathrm{op}}^2} \,.$$

*Proof.* By definition of $\mathcal{A}(D)$,

$$\frac{D^2}{T - p'} \frac{\|\Delta_{\boldsymbol{A}}\|_F^2}{\|\Delta_{\boldsymbol{A}}\|_{\mathrm{op}}^2} \geqslant \frac{1}{T - p'} \|\Delta_{\boldsymbol{A}}\|_F^2 \,,$$

and the result follows by Theorem C.12. $\square$

**Proposition C.14.** *The empirical risk minimizer $\hat{\boldsymbol{A}}$, i.e.,*

$$\hat{\boldsymbol{A}} \in \operatorname{argmin}_{\boldsymbol{A} \in \mathcal{A}(D)} \mathcal{L}(\boldsymbol{A}) \,, \tag{21}$$

*implies $\mathcal{L}(\hat{\boldsymbol{A}}) \leqslant \mathcal{L}(\boldsymbol{A}_{p'}^{\star})$, which can be rewritten as follows:*

$$\left\|\Delta_{\hat{\boldsymbol{A}}} L_{\star} E\right\|_F^2 \leqslant 2 \operatorname{Tr}\left(E^{\top} L_{\star}^{\top} \Delta_{\hat{\boldsymbol{A}}}^{\top}\left(I - M_{\boldsymbol{A}_{p'}^{\star}}\right) L_{\star} E\right) \,.$$

*Proof.* By Theorem C.8,

$$\begin{aligned}
NT\mathcal{L}(\boldsymbol{A}) &= \|\left(M_{\boldsymbol{A}} - I\right) L_{\star} E\|_2^2 = \|\left[\left(M_{\boldsymbol{A}} - M_{\boldsymbol{A}^{\star}}\right) L_{\star} - I\right] E\|_F^2 \\
&= \left\|\left[\left(M_{\boldsymbol{A}} - M_{\boldsymbol{A}_{p'}^{\star}}\right) L_{\star} + \left(M_{\boldsymbol{A}_{p'}^{\star}} - M_{\boldsymbol{A}^{\star}}\right) L_{\star} - I\right] E\right\|_F^2 \\
&= \left\|\left(M_{\boldsymbol{A}} - M_{\boldsymbol{A}_{p'}^{\star}}\right) L_{\star} E\right\|_F^2 + \left\|\left[\left(M_{\boldsymbol{A}_{p'}^{\star}} - M_{\boldsymbol{A}^{\star}}\right) L_{\star} - I\right] E\right\|_F^2 \\
&\quad + 2\operatorname{Tr}\left(E^{\top} L_{\star}^{\top}\left(M_{\boldsymbol{A}} - M_{\boldsymbol{A}_{p'}^{\star}}\right)^{\top}\left[\left(M_{\boldsymbol{A}_{p'}^{\star}} - M_{\boldsymbol{A}^{\star}}\right) L_{\star} - I\right] E\right) \\
&= \left\|\left(M_{\boldsymbol{A}} - M_{\boldsymbol{A}_{p'}^{\star}}\right) L_{\star} E\right\|_F^2 + \left\|\left(M_{\boldsymbol{A}_{p'}^{\star}} - I\right) L_{\star} E\right\|_F^2 \\
&\quad + 2\operatorname{Tr}\left(E^{\top} L_{\star}^{\top}\left(M_{\boldsymbol{A}} - M_{\boldsymbol{A}_{p'}^{\star}}\right)^{\top}\left(M_{\boldsymbol{A}_{p'}^{\star}} - I\right) L_{\star} E\right) \,.
\end{aligned}$$

Then, we have

$$
\begin{aligned}
NT\left(\mathcal{L}(\boldsymbol{A}) - \mathcal{L}(\boldsymbol{A}_{p'}^{\star})\right) &= \left\|\left(M_{\boldsymbol{A}} - M_{\boldsymbol{A}_{p'}^{\star}}\right) L_{\star} E\right\|_F^2 \\
&\quad + 2\operatorname{Tr}\left(E^\top L_{\star}^\top \left(M_{\boldsymbol{A}} - M_{\boldsymbol{A}_{p'}^{\star}}\right)^\top \left(M_{\boldsymbol{A}_{p'}^{\star}} - I\right) L_{\star} E\right),
\end{aligned}
\tag{22}
$$

which implies the desired result for any $\boldsymbol{A}$ that satisfies $\mathcal{L}(\boldsymbol{A}) \leqslant \mathcal{L}(\boldsymbol{A}_{p'}^{\star})$. □

**Corollary C.15.** *Observe that for $p' = p$, Theorem C.14 reads*

$$
\left\|\Delta_{\hat{\boldsymbol{A}}} L_{\star} E\right\|_F^2 \leqslant 2\operatorname{Tr}\left(E^\top \Delta_{\hat{\boldsymbol{A}}} L_{\star} E\right).
$$

*For $p' < p$, one can write the following relaxed condition for any $\hat{\boldsymbol{A}}$:*

$$
\begin{aligned}
\left\|\Delta_{\hat{\boldsymbol{A}}} L_{\star} E\right\|_F^2 &\leqslant 2\left\|\left(I - M_{\boldsymbol{A}_{p'}^{\star}}\right) L_{\star}\right\|_{\mathrm{op}} \operatorname{Tr}\left(E^\top \Delta_{\hat{\boldsymbol{A}}} L_{\star} E\right) \\
&= 2\left\|I_{Td} + \left(M_{\boldsymbol{A}^{\star}} - M_{\boldsymbol{A}_{p'}^{\star}}\right) L_{\star}\right\|_{\mathrm{op}} \operatorname{Tr}\left(E^\top \Delta_{\hat{\boldsymbol{A}}} L_{\star} E\right) \\
&\leqslant 2\left(1 + \left\|\left(M_{\boldsymbol{A}^{\star}} - M_{\boldsymbol{A}_{p'}^{\star}}\right) L_{\star}\right\|_{\mathrm{op}}\right) \operatorname{Tr}\left(E^\top \Delta_{\hat{\boldsymbol{A}}} L_{\star} E\right) \\
&= 2\eta \operatorname{Tr}\left(E^\top \Delta_{\hat{\boldsymbol{A}}} L_{\star} E\right).
\end{aligned}
$$

### C.3 LOWER AND UPPER ISOMETRIES

In Theorem C.16, we present uniform bounds on $\|\Delta_{\boldsymbol{A}} L_{\star} E\|_F^2$ and $\operatorname{Tr}(E^\top \Delta_{\boldsymbol{A}} L_{\star} E)$ in terms of $\|\Delta_{\boldsymbol{A}}\|_F$. In order to establish these bounds, we first start with point-wise bounds in Theorems C.19 and C.20 that rely on Hanson-Wright inequality for bounding the deviation of quadratic forms. Then, we use Theorems C.21 to C.23 with a discretization argument to establish uniform isometries. Finally, with Theorem C.17, we have a uniform control over the range of both $\|\Delta_{\boldsymbol{A}} L_{\star} E\|_F^2$ and $\operatorname{Tr}(E^\top \Delta_{\boldsymbol{A}} L_{\star} E)$.

**Theorem C.16.** *Let $\delta > 0$ be small and fixed. Then, there exists a constant $1 \leqslant C(\delta) = \mathcal{O}(\ln(1/\delta))$ such that the following holds uniformly for all $\boldsymbol{A} \in \mathcal{A}(D) \setminus \mathcal{G}(C, D)$ and $C \geqslant C(\delta)$:*

$$
\begin{aligned}
\|\Delta_{\boldsymbol{A}} L_{\star} E\|_F^2 &\geqslant \frac{\sigma^2}{8} \sigma_{\min}(L_{\star})^2 N \|\Delta_{\boldsymbol{A}}\|_F^2, \\
\operatorname{Tr}(E^\top \Delta_{\boldsymbol{A}} L_{\star} E) &\leqslant \sigma^2 \|L_{\star}\|_{\mathrm{op}} \sqrt{C\tau p' dr N} \|\Delta_{\boldsymbol{A}}\|_F,
\end{aligned}
$$

*with probability at least $1 - \delta$.*

*Proof.* Let $\nu_1, \nu_2 \in (0, 1)$ be arbitrary. By Theorems C.19 and C.20, with probability at least $1 - \delta_1 - \delta_2$, the following holds:

$$
\begin{aligned}
\|\Delta_{\boldsymbol{A}} L_{\star} E\|_F^2 &\geqslant \sigma^2 \left(1 - c^2 \nu_1\right) \sigma_{\min}(L_{\star})^2 N \|\Delta_{\boldsymbol{A}}\|_F^2, \\
\operatorname{Tr}(E^\top \Delta_{\boldsymbol{A}} L_{\star} E) &\leqslant \sigma^2 c^2 \nu_2 \|L_{\star}\|_{\mathrm{op}} \sqrt{C\eta^2 \tau p' dr N} \|\Delta_{\boldsymbol{A}}\|_F,
\end{aligned}
\tag{23}
$$

for any arbitrary $\nu_1, \nu_2 \in (0, 1)$ and $\boldsymbol{A} \in \mathcal{A}(D) \setminus \mathcal{G}(C, D)$ where

$$
\delta_1 = \exp\left(-C_{HW} C \nu_1^2 \eta^2 \tau p' dr\right), \quad \delta_2 = \exp\left(-C_{HW} C \nu_2^2 \eta^2 \tau p' dr\right).
$$

Let $\mathcal{B}(C, D)$ be the normalized $\mathcal{A}(D) \setminus \mathcal{G}(C, D)$,

$$
\mathcal{B}(C, D) = \left\{\frac{\boldsymbol{A}}{\|\boldsymbol{A}\|_F} \mid \boldsymbol{A} \in \mathcal{A}(D) \setminus \mathcal{G}(C, D)\right\}.
$$

Then, since the conditions are homogeneous, Equation (23) holds for any $\boldsymbol{A} \in \mathcal{B}(C, D)$ with probability $1 - \delta_1 - \delta_2$.

Let $\mathcal{N}_\epsilon(D)$ be $\epsilon$-net over the set $\mathcal{B}(C, D)$. Hence, with probability at least

$$
1 - \delta_0 = 1 - |\mathcal{N}_\epsilon(D)|(\delta_1 + \delta_2),
$$

the condition Equation (23) holds $\forall \boldsymbol{A} \in \mathcal{N}_\epsilon(D)$. Moreover, by Theorems C.21 to C.23, we have

$$\|\Delta_{\boldsymbol{A}} L_\star E\|_F^2 \geqslant \frac{1}{2}\sigma^2 \left(1 - c^2\nu_1\right) \sigma_{\min}(L_\star)^2 N\|\Delta_{\boldsymbol{A}}\|_F^2 - \sigma^2(1 + c^2\nu_3)\epsilon^2\|L_\star\|_{\mathrm{op}}^2 p'dNT$$

$$\mathrm{Tr}(E^\top \Delta_{\boldsymbol{A}} L_\star E) \leqslant \sigma^2 c^2\nu_2\|L_\star\|_{\mathrm{op}}\sqrt{C\eta^2\tau p'drN}\|\Delta_{\boldsymbol{A}}\|_F + \sigma^2(1 + c^2\nu_3)\epsilon\|L_\star\|_{\mathrm{op}}\sqrt{p'dNT},$$

$\forall \boldsymbol{A} \in \mathcal{B}(C, D)$ with probability at least $1 - \delta_0 - \delta_3$ where

$$\delta_3 = \exp\left(-C_{HW}\min\left\{\nu_3, \nu_3^2\right\}dNT\right).$$

Recall that

$$\|\Delta_{\boldsymbol{A}}\|_F^2 \geqslant (T - p)\|\boldsymbol{A}\|_F^2 = T - p,$$

for any $\boldsymbol{A} \in \mathcal{B}(C, D)$ due to the normalization.

Setting $\nu_1 = \frac{1}{4c^2}, \nu_2 = \frac{1}{2c^2}$ and $\epsilon$ such that

$$\epsilon = \frac{1}{2\left(1 + c^2\nu_3\right)}\sqrt{\frac{T - p'}{T}}\min\left\{\frac{1}{\sigma_{\mathrm{cond}}(L_\star)}\sqrt{\frac{1}{p'd}}, \sqrt{\frac{C\eta^2\tau r}{dNT}}\right\},$$

and we have the following:

$$\frac{1}{2}\sigma^2 \left(1 - c^2\nu_1\right) \sigma_{\min}(L_\star)^2 N\|\Delta_{\boldsymbol{A}}\|_F^2 - \sigma^2(1 + c^2\nu_3)\epsilon^2\|L_\star\|_{\mathrm{op}}^2 p'dNT \geqslant \frac{\sigma^2}{8}\sigma_{\min}(L_\star)^2 N\|\Delta_{\boldsymbol{A}}\|_F^2,$$

$$\sigma^2 c^2\nu_2\|L_\star\|_{\mathrm{op}}\sqrt{C\eta^2\tau p'drN}\|\Delta_{\boldsymbol{A}}\|_F + \sigma^2(1 + c^2\nu_3)\epsilon\|L_\star\|_{\mathrm{op}}\sqrt{p'dNT}$$
$$\leqslant \sigma^2\|L_\star\|_{\mathrm{op}}\sqrt{C\eta^2\tau p'drN}\|\Delta_{\boldsymbol{A}}\|_F,$$

$\forall \boldsymbol{A} \in \mathcal{B}(C, D)$ with probability $1 - \delta_0 - \delta_3$.

By homogeneity, this implies that $\forall \boldsymbol{A} \in \mathcal{A}(D) \setminus \mathcal{G}(C, D)$, with probability at least $1 - \delta_0 - \delta_3$,

$$\|\Delta_{\boldsymbol{A}} L_\star E\|_F^2 \geqslant \frac{\sigma^2}{8}\sigma_{\min}(L_\star)^2 N\|\Delta_{\boldsymbol{A}}\|_F^2,$$

$$\mathrm{Tr}(E^\top \Delta_{\boldsymbol{A}} L_\star E) \leqslant \sigma^2\|L_\star\|_{\mathrm{op}}\sqrt{C\eta^2\tau p'drN}\|\Delta_{\boldsymbol{A}}\|_F.$$

Lastly, we need ensure that $\delta_0 + \delta_3 \leqslant \delta$. First, $\delta_3 \leqslant \delta/2$ can be achieved with the choice of

$$\nu_3(\delta/2) = \max\left\{1, \frac{\ln\frac{1}{\delta/2}}{C_{HW}dNT}\right\} = \mathcal{O}\left(\ln(1/\delta)\right).$$

Moreover, the $\epsilon$-net size is bounded as follows:

$$|\mathcal{N}_\varepsilon(D)| \leqslant \exp\left(9p'dr\ln\frac{p'}{\epsilon}\right),$$

by Theorem B.9. Then,

$$\delta_0 = |\mathcal{N}_\epsilon(D)|\left(\delta_1 + \delta_2\right)$$
$$\leqslant \exp\left(-\frac{5}{16c^4}C_{HW}C\eta^2\tau p'dr + 9p'dr\ln\frac{p'}{\epsilon}\right).$$

Thus, $\delta_0 \leqslant \delta/2$ can be achieved with the choice of

$$C \geqslant C(\delta) = \frac{16c^4}{5C_{HW}\eta^2\tau}\left(9\ln\frac{p'}{\epsilon} + \frac{\ln 2/\delta}{p'dr}\right). \tag{24}$$

Note that $\eta, \tau \geqslant 1$ and the latter term is $\mathcal{O}(\ln(1/\delta))$. For the first term, crudely upper bound $\ln\frac{p'}{\epsilon}$ by using $x \geqslant \ln x + 1$ for any $x \geqslant 0$:

$$\ln\frac{p'}{\epsilon} \leqslant \ln\sqrt{\frac{p'^2dNT^2}{T - p'}} + \ln\sigma_{\mathrm{cond}}(L_\star) + \ln 2(1 + c^2\nu_3(\delta/2)) \leqslant \tau + \mathcal{O}(\ln(1/\delta)).$$

Therefore, the definition in Equation (24) verifies $C(\delta) = \mathcal{O}(\ln(1/\delta))$.

$\square$

**Corollary C.17.** *For any small $\delta > 0$, there exists a constant $1 \leqslant C(\delta) = \mathcal{O}(\ln(1/\delta))$ such that the following holds uniformly for all $\boldsymbol{A} \in \mathcal{A}(D) \setminus \mathcal{G}(C(\delta), D)$ and $C \geqslant C(\delta)$:*

$$\inf_{\boldsymbol{A} \in \mathcal{A}(D) \setminus \mathcal{G}(C,D)} \|\Delta_{\boldsymbol{A}} L_\star E\|_F^2 \geqslant \frac{\sigma^2}{8} \sigma_{\min}(L_\star)^2 C(\delta) D^2 \eta^2 \tau p' dr \,,$$

$$\sup_{\boldsymbol{A} \in \mathcal{A}(D) \setminus \mathcal{G}(C,D)} \mathrm{Tr}(E^\top \Delta_{\boldsymbol{A}} L_\star E) \leqslant \sigma^2 \|L_\star\|_{\mathrm{op}} \sqrt{C(\delta) D^2 \eta^2 \tau p' d^2 r N T} \,.$$

*Proof.* Plug in the results from Theorem C.16 and use the definition of $\mathcal{A}(D) \setminus \mathcal{G}(C(\delta), D)$ to lower and upper bound $\|\Delta_{\boldsymbol{A}}\|_F$. $\qquad\square$

**Definition C.18.** *For applying Theorem B.1 in our setup, consider the following objects:*

$$\tilde{E} = (\xi^{(1)^\top}, \ldots, \xi^{(N)^\top})^\top \in \mathbb{R}^{NTd} \,,$$

$$\tilde{\Delta}_{\boldsymbol{A}} = \mathrm{diag}(\Delta_{\boldsymbol{A}}) \in \mathbb{R}^{NTd} \times \mathbb{R}^{NTd} \,,$$

*where $\mathrm{diag}(P)$ puts $P$ in the diagonal blocks of a larger diagonal matrix.*

**Lemma C.19.** *For any $\boldsymbol{A} \in \mathcal{A} \setminus \mathcal{G}(C)$ and $\nu \in (0, 1)$, with probability at least*

$$1 - \exp\left(-C_{HW} C \nu^2 \eta^2 \tau p' dr\right) \,,$$

*we have the following*

$$\|\Delta_{\boldsymbol{A}} L_\star E\|_F^2 \geqslant \sigma^2 \left(1 - c^2 \nu\right) \sigma_{\min}(L_\star)^2 N \|\Delta_{\boldsymbol{A}}\|_F^2.$$

*Proof.* First, observe that

$$\|\Delta_{\boldsymbol{A}} L_\star E\|_F^2 \geqslant \sigma_{\min}(L_\star)^2 \|\Delta_{\boldsymbol{A}} E\|_F^2 \,.$$

Applying Theorem B.1 with $P = \tilde{\Delta}_{\boldsymbol{A}}^\top \tilde{\Delta}_{\boldsymbol{A}}$ and $r = c^2 \sigma^2 \nu \|\tilde{\Delta}_{\boldsymbol{A}}\|_F^2$,

$$\mathbb{P}\left(\tilde{E}^\top \tilde{\Delta}_{\boldsymbol{A}}^\top \tilde{\Delta}_{\boldsymbol{A}} \tilde{E} - \mathbb{E}[\tilde{E}^\top \tilde{\Delta}_{\boldsymbol{A}}^\top \tilde{\Delta}_{\boldsymbol{A}} \tilde{E}] \geqslant c^2 \sigma^2 \nu \|\tilde{\Delta}_{\boldsymbol{A}}\|_F^2\right)$$

$$\leqslant \exp\left(-C_{HW} \min\left\{\nu^2 \frac{\|\tilde{\Delta}_{\boldsymbol{A}}\|_F^4}{\|\tilde{\Delta}_{\boldsymbol{A}}^\top \tilde{\Delta}_{\boldsymbol{A}}\|_F^2}, \nu \frac{\|\tilde{\Delta}_{\boldsymbol{A}}\|_F^2}{\|\tilde{\Delta}_{\boldsymbol{A}}^\top \tilde{\Delta}_{\boldsymbol{A}}\|_{\mathrm{op}}}\right\}\right) \,.$$

Observe that $\tilde{E}^\top \tilde{\Delta}_{\boldsymbol{A}}^\top \tilde{\Delta}_{\boldsymbol{A}} \tilde{E} = \mathrm{Tr}(E^\top \Delta_{\boldsymbol{A}}^\top \Delta_{\boldsymbol{A}} E) = \|\Delta_{\boldsymbol{A}} E\|_F^2$ and

$$\mathbb{E}\left[\tilde{E}^\top \tilde{\Delta}_{\boldsymbol{A}}^\top \tilde{\Delta}_{\boldsymbol{A}} \tilde{E}\right] = \mathbb{E}\left[\mathrm{Tr}\left(\tilde{E}\tilde{E}^\top \tilde{\Delta}_{\boldsymbol{A}}^\top \tilde{\Delta}_{\boldsymbol{A}}\right)\right] = \sigma^2 \|\tilde{\Delta}_{\boldsymbol{A}}\|_F^2 = \sigma^2 N \|\Delta_{\boldsymbol{A}}\|_F^2.$$

Furthermore, $\|\tilde{\Delta}_{\boldsymbol{A}}\|_F^4 = N^2 \|\Delta_{\boldsymbol{A}}\|_F^4, \|\tilde{\Delta}_{\boldsymbol{A}}^\top \tilde{\Delta}_{\boldsymbol{A}}\|_F^2 = N \|\Delta_{\boldsymbol{A}}^\top \Delta_{\boldsymbol{A}}\|_F^2$ and $\|\tilde{\Delta}_{\boldsymbol{A}}^\top \tilde{\Delta}_{\boldsymbol{A}}\|_{\mathrm{op}} = \|\Delta_{\boldsymbol{A}}^\top \Delta_{\boldsymbol{A}}\|_{\mathrm{op}} = \|\Delta_{\boldsymbol{A}}\|_{\mathrm{op}}^2$. Plugging these into the bound,

$$\mathbb{P}\left(\|\Delta_{\boldsymbol{A}} E\|_F^2 - \sigma^2 N \|\Delta_{\boldsymbol{A}}\|_F^2 \geqslant c^2 \sigma^2 \nu N \|\Delta_{\boldsymbol{A}}\|_F^2\right) \leqslant$$

$$\exp\left(-C_{HW} \min\left\{\nu^2 N \frac{\|\Delta_{\boldsymbol{A}}\|_F^4}{\|\Delta_{\boldsymbol{A}}^\top \Delta_{\boldsymbol{A}}\|_F^2}, \nu N \frac{\|\Delta_{\boldsymbol{A}}\|_F^2}{\|\Delta_{\boldsymbol{A}}\|_{\mathrm{op}}^2}\right\}\right) \,.$$

Then, using $\|\Delta_{\boldsymbol{A}}^\top \Delta_{\boldsymbol{A}}\|_F^2 \leqslant \|\Delta_{\boldsymbol{A}}\|_F^2 \|\Delta_{\boldsymbol{A}}\|_{\mathrm{op}}^2$ and $\nu < 1$,

$$\mathbb{P}\left(\|\Delta_{\boldsymbol{A}} E\|_F^2 \geqslant \sigma^2 (1 - c^2 \nu) N \|\Delta_{\boldsymbol{A}}\|_F^2\right) \geqslant 1 - \exp\left(-C_{HW} \nu^2 N \frac{\|\Delta_{\boldsymbol{A}}\|_F^2}{\|\Delta_{\boldsymbol{A}}\|_{\mathrm{op}}^2}\right) \,.$$

The result follows from the definition of set $\mathcal{A} \setminus \mathcal{G}(C)$. $\qquad\square$

**Lemma C.20.** *For any $\boldsymbol{A} \in \mathcal{A} \setminus \mathcal{G}(C)$ and $\nu \in (0, 1)$, with probability at least*

$$1 - \exp\left(-C_{HW} C \nu^2 \eta^2 \tau p' dr\right) \,,$$

*we have the following*

$$\mathrm{Tr}(E^\top \Delta_{\boldsymbol{A}} L_\star E) \leqslant c^2 \sigma^2 \nu \|L_\star\|_{\mathrm{op}} \sqrt{C \eta^2 \tau p' dr N} \|\Delta_{\boldsymbol{A}}\|_F \,.$$

*Proof.* First, by the properties of trace and Frobenius norm, we have

$$\text{Tr}(E^\top \Delta_{\boldsymbol{A}} L_\star E) \leqslant \|L_\star\|_{\text{op}} \text{Tr}(E^\top \Delta_{\boldsymbol{A}} E).$$

Applying Theorem B.1 with $P = \tilde{\Delta}_{\boldsymbol{A}}$ and $r = c^2\sigma^2\nu\sqrt{C\eta^2\tau p'dr}\|\tilde{\Delta}_{\boldsymbol{A}}\|_F$,

$$\mathbb{P}\left(\tilde{E}^\top \tilde{\Delta}_{\boldsymbol{A}} \tilde{E} - \mathbb{E}[\tilde{E}^\top \tilde{\Delta}_{\boldsymbol{A}} \tilde{E}] \geqslant c^2\sigma^2\nu\sqrt{C\eta^2\tau p'dr}\|\tilde{\Delta}_{\boldsymbol{A}}\|_F\right)$$
$$\leqslant \exp\left(-C_{HW}\min\left\{\nu^2 C\eta^2\tau p'dr, \nu\sqrt{C\eta^2\tau p'dr}\frac{\|\tilde{\Delta}_{\boldsymbol{A}}\|_F}{\|\tilde{\Delta}_{\boldsymbol{A}}\|_{\text{op}}}\right\}\right).$$

Noting $\mathbb{E}[\tilde{E}^\top \tilde{\Delta}_{\boldsymbol{A}} \tilde{E}] = 0$ and rewriting,

$$\mathbb{P}\left(\text{Tr}\left(E^\top \Delta_{\boldsymbol{A}} E\right) \leqslant c^2\sigma^2\nu\sqrt{C\eta^2\tau p'drN}\|\Delta_{\boldsymbol{A}}\|_F\right) \geqslant$$
$$1 - \exp\left(-C_{HW}\min\left\{\nu^2 C\eta^2\tau p'dr, \nu\sqrt{C\eta^2\tau p'drN}\frac{\|\Delta_{\boldsymbol{A}}\|_F}{\|\Delta_{\boldsymbol{A}}\|_{\text{op}}}\right\}\right).$$

The result follows from the definition of set $\mathcal{A} \setminus \mathcal{G}(C)$. $\qquad\square$

**Lemma C.21.** *For any $\nu > 0$, with probability at least*

$$1 - \exp\left(-C_{HW}\min\left\{\nu, \nu^2\right\}dNT\right),$$

*we have the following*

$$\|E\|_F^2 - \sigma^2 dNT \leqslant c^2\sigma^2\nu dNT.$$

*Proof.* Applying Theorem B.1 with $P = I_{dTN}, r = c^2\sigma^2\nu dNT$,

$$\mathbb{P}\left(\tilde{E}^\top \tilde{E} - \mathbb{E}[\tilde{E}^\top \tilde{E}] \geqslant c^2\sigma^2\nu dNT\right) \leqslant \exp\left(-C_{HW}\nu dNT\right).$$

The result follows by the fact $\mathbb{E}[\tilde{E}^\top \tilde{E}] = \sigma^2 dNT$. $\qquad\square$

**Lemma C.22.** *For any $\boldsymbol{A}_1, \boldsymbol{A}_2 \in \mathcal{A}$,*

$$\|\Delta_{\boldsymbol{A}_2} L_\star E\|_F^2 \geqslant \frac{1}{2}\|\Delta_{\boldsymbol{A}_1} L_\star E\|_F^2 - p'\|\boldsymbol{A}_1 - \boldsymbol{A}_2\|_{\text{op}}^2\|L_\star\|_{\text{op}}^2\|E\|_F^2.$$

*Proof.* By the properties of Frobenius norm and Theorem C.12,

$$\|\Delta_{\boldsymbol{A}_1} L_\star E\|_F^2 = \|(\Delta_{\boldsymbol{A}_1} - \Delta_{\boldsymbol{A}_2} + \Delta_{\boldsymbol{A}_2})L_\star E\|_F^2$$
$$\leqslant 2\|\Delta_{\boldsymbol{A}_2} L_\star E\|_F^2 + 2\|(\Delta_{\boldsymbol{A}_1} - \Delta_{\boldsymbol{A}_2})L_\star E\|_F^2$$
$$\leqslant 2\|\Delta_{\boldsymbol{A}_2} L_\star E\|_F^2 + 2\|\Delta_{\boldsymbol{A}_1} - \Delta_{\boldsymbol{A}_2}\|_{\text{op}}^2\|L_\star\|_{\text{op}}^2\|E\|_F^2$$
$$\leqslant 2\|\Delta_{\boldsymbol{A}_2} L_\star E\|_F^2 + 2p'\|\boldsymbol{A}_1 - \boldsymbol{A}_2\|_{\text{op}}^2\|L_\star\|_{\text{op}}^2\|E\|_F^2.$$

The results readily follows by reordering terms. $\qquad\square$

**Lemma C.23.** *For any $\boldsymbol{A}_1, \boldsymbol{A}_2 \in \mathcal{A}$,*

$$\text{Tr}(E^\top \Delta_{\boldsymbol{A}_2} L_\star E) \leqslant \text{Tr}(E^\top \Delta_{\boldsymbol{A}_1} L_\star E) + \sqrt{p'}\|\boldsymbol{A}_1 - \boldsymbol{A}_2\|_{\text{op}}\|L_\star\|_{\text{op}}\|E\|_F^2,$$

*Proof.* By the properties of trace and Theorem C.12,

$$\text{Tr}(E^\top \Delta_{\boldsymbol{A}_2} L_\star E) = \text{Tr}\left(E^\top (\Delta_{\boldsymbol{A}_2} - \Delta_{\boldsymbol{A}_1} + \Delta_{\boldsymbol{A}_1})L_\star E\right)$$
$$= \text{Tr}\left(E^\top \Delta_{\boldsymbol{A}_1} L_\star E\right) + \text{Tr}\left(E^\top (\Delta_{\boldsymbol{A}_2} - \Delta_{\boldsymbol{A}_1})L_\star E\right)$$
$$\leqslant \text{Tr}\left(E^\top \Delta_{\boldsymbol{A}_1} L_\star E\right) + \|\Delta_{\boldsymbol{A}_2} - \Delta_{\boldsymbol{A}_1}\|_{\text{op}}\|L_\star\|_{\text{op}}\|E\|_F^2$$
$$\leqslant \text{Tr}\left(E^\top \Delta_{\boldsymbol{A}_1} L_\star E\right) + \sqrt{p'}\|\boldsymbol{A}_1 - \boldsymbol{A}_2\|_{\text{op}}\|L_\star\|_{\text{op}}\|E\|_F^2.$$

$\qquad\square$

### C.4 CONCENTRATION INEQUALITIES

In Theorem C.24, we show that the quantities of interest that show up in Equation (19) are related to a martingale series and its predictable quadratic variation. This allow us to use Theorem B.5 in Theorem C.26 to quantify the probability of the event in Equation (19) for finite sets of $\boldsymbol{A}$.

**Remark C.24.** *Fix $\boldsymbol{A} \in \mathcal{A}(D)$. Consider the martingale differences sequences*

$$d_{t,i}^{(n)} = \left( \left( \boldsymbol{A} - \boldsymbol{A}_{p'}^\star \right) X_t^{(n)} \right)_i \left( \xi_t^{(n)} \right)_i ,$$

*where the series is first ordered in $i$, then in $t$, and finally in $n$. Let $Y$ be the sum of the martingale differences, i.e.,*

$$Y = \sum_{i,t,n} d_{i,t}^{(n)} .$$

*Let $W_{\boldsymbol{A}}^R$ be the quadratic variation of the series plus an error term as in Theorem B.4, i.e.,*

$$W_{\boldsymbol{A}}^R = \sum_{i,t,n} \mathbb{E}_{\left( \xi_t^{(n)} \right)_i} \left[ \left( d_{i,t}^{(n)} \right)^2 \right] + \sum_{n,t,i} \mathbb{1}_{d_{t,i}^{(n)} > R} \left( d_{t,i}^{(n)} \right)^2 ,$$

*Then, we have the following computations:*

$$Y_{\boldsymbol{A}} = \sum_{n,t} \langle (\boldsymbol{A} - \boldsymbol{A}_{p'}^\star) X_t^{(n)}, \xi_t^{(n)} \rangle = \mathrm{Tr}\left( E^\top \Delta_{\boldsymbol{A}} L_\star E \right) ,$$

$$W_{\boldsymbol{A}} \coloneqq W_{\boldsymbol{A}}^0 = \sigma^2 \sum_{n,t} \left\| \left( \boldsymbol{A} - \boldsymbol{A}_{p'}^\star \right) X_t^{(n)} \right\|^2 = \sigma^2 \| \Delta_{\boldsymbol{A}} L_\star E \|_F^2 .$$

**Proposition C.25.** *Let $0 < \delta < e^{-1}$ and $R > 0$ be constants. Then, with probability at least $1 - \delta$, we have*

$$\forall \boldsymbol{A} \in \mathcal{A}(D), \quad C' \left( \ln 2dNT + \ln \frac{1}{\delta} \right) W_{\boldsymbol{A}} \geqslant W_{\boldsymbol{A}}^R , \tag{25}$$

*where $C' = 1 + 4c'^2$ and $c'$ is the universal constant in Theorem B.6.*

*Proof.* By Theorem B.7, there exists a constant $c'$ such that

$$\sup_{t,n} \|\xi_t^{(n)}\|_\infty \leqslant c'\sigma\sqrt{2} \left( \sqrt{\ln 2dTN} + \sqrt{\ln \frac{1}{\delta}} \right) .$$

Therefore, for any $\boldsymbol{A} \in \mathcal{A}(D)$, we have

$$\begin{aligned}
W_{\boldsymbol{A}}^R &= W_{\boldsymbol{A}} + \sum_{n,t,i} \mathbb{1}_{d_{t,i}^{(n)} > R} \left( d_{t,i}^{(n)} \right)^2 \\
&\leqslant W_{\boldsymbol{A}} + 4c'^2\sigma^2 \left( \ln 2dTN + \ln \frac{1}{\delta} \right) \sum_{n,t,i} \mathbb{1}_{d_{t,i}^{(n)} > R} \left( \left( \boldsymbol{A} - \boldsymbol{A}_{p'}^\star \right) X_t^{(n)} \right)_i^2 \\
&\leqslant W_{\boldsymbol{A}} + 4c'^2\sigma^2 \left( \ln 2dTN + \ln \frac{1}{\delta} \right) \sum_{n,t,i} \left( \left( \boldsymbol{A} - \boldsymbol{A}_{p'}^\star \right) X_t^{(n)} \right)_i^2 \\
&\leqslant W_{\boldsymbol{A}} + 4c'^2 \left( \ln 2dTN + \ln \frac{1}{\delta} \right) W_{\boldsymbol{A}} .
\end{aligned}$$

Then, by rearranging terms, we have

$$\left( 1 + 4c'^2 \left( \ln 2dTN + \ln \frac{1}{\delta} \right) \right) W_{\boldsymbol{A}} \geqslant W_{\boldsymbol{A}}^R .$$

$\square$

**Theorem C.26.** *Let $\mathcal{S} \subseteq \mathcal{A}(D)$ be a finite set and let $\mathcal{E}(\mathcal{S})$ be the following event*

$$\mathcal{E}(\mathcal{S}) = \left\{ \inf_{\boldsymbol{A} \in \mathcal{S}} W_{\boldsymbol{A}} \geqslant \alpha_L \right\} \cap \left\{ \sup_{\boldsymbol{A} \in \mathcal{S}} Y_{\boldsymbol{A}} \leqslant \alpha_U \right\},$$

*where $\alpha_L, \alpha_U > 0$ are two constants. Then, for any $\gamma, R > 0$ and $0 < \delta < e^{-1}$,*

$$\mathbb{P}\left( \{ \exists \boldsymbol{A} \in \mathcal{S} : W_{\boldsymbol{A}} \leqslant \gamma Y_{\boldsymbol{A}} \} \cap \mathcal{E}(\mathcal{S}) \right)$$

$$\leqslant |\mathcal{S}| \exp \left( -\frac{\alpha_L}{2\gamma'(\gamma' + R)} + \ln \left( \ln \left( \frac{\gamma' \alpha_U}{\alpha_L} \right) + 1 \right) \right) + \delta,$$

*where $\gamma' = C'e\gamma \left( \ln 2dTN + \ln \frac{1}{\delta} \right)$ and $C'$ is the universal constant in Theorem C.25.*

*Proof.* For any $\boldsymbol{A}$, let $\mathcal{E}_{\boldsymbol{A}}$ be the following event:

$$\mathcal{E}_{\boldsymbol{A}} = \left\{ W_{\boldsymbol{A}}^R \geqslant \alpha_L \right\} \cap \{ Y_{\boldsymbol{A}} \leqslant \alpha_U \}.$$

Then, by union bound, we have

$$\mathbb{P}\left( \{ \exists \boldsymbol{A} \in \mathcal{S} : W_{\boldsymbol{A}}^R \leqslant \gamma Y_{\boldsymbol{A}} \} \cap \mathcal{E}(\mathcal{S}) \right) \leqslant \sum_{\boldsymbol{A} \in \mathcal{S}} \mathbb{P}\left( \{ W_{\boldsymbol{A}}^R \leqslant \gamma Y_{\boldsymbol{A}} \} \cap \mathcal{E}(\mathcal{S}) \right)$$

$$\leqslant \sum_{\boldsymbol{A} \in \mathcal{S}} \mathbb{P}\left( \{ W_{\boldsymbol{A}}^R \leqslant \gamma Y_{\boldsymbol{A}} \} \cap \mathcal{E}_{\boldsymbol{A}} \right).$$

For any $\boldsymbol{A} \in \mathcal{S}$,

$$\mathbb{P}\left( W_{\boldsymbol{A}}^R \leqslant \gamma Y_{\boldsymbol{A}} \cap \mathcal{E}_{\boldsymbol{A}} \right) \leqslant \exp \left( -\frac{\alpha_L}{2e\gamma(e\gamma + R)} + \ln \left( \ln \left( \frac{\gamma \alpha_U}{\alpha_L} \right) + 1 \right) \right).$$

by Theorem B.5 which implies that

$$\mathbb{P}\left( \{ \exists \boldsymbol{A} \in \mathcal{S} : W_{\boldsymbol{A}}^R \leqslant \gamma Y_{\boldsymbol{A}} \} \cap \mathcal{E}(\mathcal{S}) \right) \leqslant |\mathcal{S}| \exp \left( -\frac{\alpha_L}{2e\gamma(e\gamma + R)} + \ln \left( \ln \left( \frac{\gamma \alpha_U}{\alpha_L} \right) + 1 \right) \right).$$

Finally, let $\mathcal{E}_\delta$ be the event in Equation (25). Then,

$$\mathbb{P}\left( \{ \exists \boldsymbol{A} \in \mathcal{S} : W_{\boldsymbol{A}} \leqslant \gamma Y_{\boldsymbol{A}} \} \cap \mathcal{E}(\mathcal{S}) \right)$$

$$\leqslant \mathbb{P}\left( \{ \exists \boldsymbol{A} \in \mathcal{S} : W_{\boldsymbol{A}} \leqslant \gamma Y_{\boldsymbol{A}} \} \cap \mathcal{E}(\mathcal{S}) \cap \mathcal{E}_\delta \right) + \mathbb{P}\left( \mathcal{E}_\delta^C \right)$$

$$\leqslant \mathbb{P}\left( \left\{ \exists \boldsymbol{A} \in \mathcal{S} : W_{\boldsymbol{A}}^R \leqslant C'\gamma \left( \ln 2dNT + \ln \frac{1}{\delta} \right) Y_{\boldsymbol{A}} \right\} \cap \mathcal{E}(\mathcal{S}) \cap \mathcal{E}_\delta \right) + \delta$$

$$\leqslant \mathbb{P}\left( \left\{ \exists \boldsymbol{A} \in \mathcal{S} : W_{\boldsymbol{A}}^R \leqslant C'\gamma \left( \ln 2dNT + \ln \frac{1}{\delta} \right) Y_{\boldsymbol{A}} \right\} \cap \mathcal{E}(\mathcal{S}) \right) + \delta.$$

$\square$

## C.5 PROOF OF THEOREM C.5

Recall that Theorem C.24 shows that

$$Y_{\boldsymbol{A}} = \operatorname{Tr}\left( E^\top \Delta_{\boldsymbol{A}} L_\star E \right), \quad W_{\boldsymbol{A}} = \sigma^2 \| \Delta_{\boldsymbol{A}} L_\star E \|_F^2.$$

Therefore, we need to show that for any $0 < \delta < e^{-1}$, there exists a constant $C(\delta)$ such that

$$\mathbb{P}\left( \exists \boldsymbol{A} \in \mathcal{A}(D) \setminus \mathcal{G}(C(\delta), D) : W_{\boldsymbol{A}} \leqslant 2\sigma^2 \eta Y_{\boldsymbol{A}} \right) \leqslant \delta.$$

By Theorem C.17, there exists a constant $C_1(\delta/4) = \mathcal{O}(\ln(1/\delta))$ such that for all $C \geqslant C_1(\delta/4)$

$$\inf_{\boldsymbol{A} \in \mathcal{A}(D) \setminus \mathcal{G}(C,D)} W_{\boldsymbol{A}} \geqslant \frac{\sigma^4}{8} \sigma_{\min}(L_\star)^2 CD^2 \eta^2 \tau p' dr,$$

$$\sup_{\boldsymbol{A} \in \mathcal{A}(D) \setminus \mathcal{G}(C,D)} Y_{\boldsymbol{A}} \leqslant \sigma^2 \| L_\star \|_{\mathrm{op}} \sqrt{CD^2 \eta^2 \tau p' d^2 r NT},$$

(26)

with probability $1 - \delta/4$. In addition, by Theorem C.21, with probability at least $1 - \delta/4$, we have

$$\|E\|_F^2 \leqslant \left(1 + c^2 \nu(\delta/4)\right) dNT\,, \quad \text{where} \quad \nu(\delta/4) = \max\left\{1, \frac{\ln\frac{\delta}{4}}{C_{HW}dNT}\right\}. \qquad (27)$$

In the following, we work conditionally on these events.

Let $\mathcal{S}$ be an $\epsilon$-net over $\mathcal{A}(D) \backslash \mathcal{G}(C, D)$. By Theorems C.22 and C.23, for any $A \in \mathcal{A}(D) \backslash \mathcal{G}(C, D)$, there exists $A' \in \mathcal{S}$ such that

$$W_A \geqslant \frac{1}{2}W_{A'} - \epsilon_1(\delta/4)\epsilon^2\,, \quad Y_A \leqslant Y_{A'} + \epsilon_2(\delta/4)\epsilon\,,$$

where $\epsilon_1(\delta/4), \epsilon_2(\delta/4)$ are given as follows:

$$\epsilon_1(\delta/4) = \sigma^4(1 + c^2\nu_3(\delta/4))p'dNT\|L_\star\|_{\text{op}}^2\,,$$
$$\epsilon_2(\delta/4) = \sigma^2(1 + c^2\nu_3(\delta/4))\sqrt{p'dNT}\|L_\star\|_{\text{op}}\,.$$

We set $\epsilon$ as follows:

$$\epsilon = \min\left\{\sqrt{\frac{\alpha_L}{\epsilon_1(\delta/4)}}, \frac{\alpha_L}{4\sigma^2\eta\epsilon_2(\delta/4)}, D\right\}.$$

In particular, $\epsilon$ is small such that for any $A \in \mathcal{A}(D) \setminus \mathcal{G}(C, D)$,

$$\exists A' \in \mathcal{S}: \quad W_{A'} \leqslant 2W_A + 2\alpha_L\,, \quad Y_{A'} \geqslant Y_A - \frac{\alpha_L}{4\sigma^2\eta}\,. \qquad (28)$$

Assume that $A \in \mathcal{A}(D) \setminus \mathcal{G}(C, D)$ verifies

$$W_A \leqslant 2\sigma^2\eta Y_A\,.$$

Then, by Equation (28), there exists $A' \in \mathcal{S}$ such that

$$W_{A'} \leqslant 4W_A \leqslant 8\sigma^2\eta Y_A \leqslant 16\sigma^2\eta Y_A - 4W_A \leqslant 16\sigma^2\eta Y_{A'}\,.$$

Therefore, we have

$$\mathbb{P}\left(\left\{\exists A \in \mathcal{A}(D) \setminus \mathcal{G}(C, D): W_A \leqslant 2\sigma^2\eta Y_A\right\} \cap \mathcal{E}\right) \leqslant \mathbb{P}\left(\left\{\exists A \in \mathcal{S}: W_A \leqslant 16\sigma^2\eta Y_A\right\} \cap \mathcal{E}\right)\,,$$

where $\mathcal{E}$ is the event that both Equation (26) and Equation (27) hold.

By Theorem C.26, there exists a universal constant $C'$ such that

$$\mathbb{P}\left(\left\{\exists A \in \mathcal{S}: W_A \leqslant 16\sigma^2\eta Y_A\right\} \cap \mathcal{E}\right) \leqslant$$
$$|\mathcal{S}| \exp\left(-\frac{\alpha_L}{4\gamma'^2} + \ln\left(\ln\left(\frac{\gamma'\alpha_U}{\alpha_L}\right) + 1\right)\right) + \frac{\delta}{4}\,, \qquad (29)$$

with the following choices:

$$\gamma = 16\sigma^2\eta\,, \quad R = \gamma' = C'e\gamma\left(\ln 2dTN + \ln\frac{1}{\delta}\right)\,,$$
$$\alpha_L = \frac{\sigma^4}{8}\sigma_{\min}(L_\star)^2CD^2\eta^2\tau p'dr\,,$$
$$\alpha_U = \sigma^2\|L_\star\|_{\text{op}}\sqrt{CD^2\eta^2\tau p'd^2rNT}\,.$$

By a union bound, Equation (29) implies that

$$\mathbb{P}\left(\exists A \in \mathcal{A}(D) \setminus \mathcal{G}(C, D): W_A \leqslant 2\sigma^2\eta Y_A\right) \leqslant$$
$$|\mathcal{S}| \exp\left(-\frac{\alpha_L}{4\gamma'^2} + \ln\left(\ln\left(\frac{\gamma'\alpha_U}{\alpha_L}\right) + 1\right)\right) + \frac{3\delta}{4}\,. \qquad (30)$$

By Theorem C.12, we have

$$\|\Delta_A\|_{\text{op}} = \|M_{A-A^\star}\|_{\text{op}} \geqslant \|A - A^\star\|_{\text{op}}\,.$$

Therefore, an $\epsilon$-net covering of $\mathcal{M}_{r,p'}(D)$ defined in Theorem B.9 is also an $\epsilon$-net over $\mathcal{A}(D)$ after a shift by $\boldsymbol{A}^{\star}$ and we have

$$\ln |\mathcal{S}| \leqslant 9p'dr \ln \frac{Dp'}{\epsilon} \,.$$

Finally, we have to show that the right-hand side of Equation (30) is upper bounded by $\delta$. That is, we need to prove

$$\frac{\alpha_L}{4\gamma'^2} \geqslant \ln \left( \ln \left( \frac{\gamma'\alpha_U}{\alpha_L} \right) + 1 \right) + 9p'dr \ln \frac{Dp'}{\epsilon} + \ln \frac{4}{\delta} \,, \tag{31}$$

for a suitable choice of $C$. Note that Equation (31) is homogeneous in $\sigma$. In addition, the left-hand side does not depend on $\eta$, whereas the right-hand side is decreasing in $\eta$. Therefore, for simplicity, we can set $\sigma = 1$ and $\eta = 1$.

By Theorem C.11 we have

$$\sigma_{\min}(L_\star) \geqslant \frac{1}{D+1} \geqslant \frac{1}{2D} \,,$$

and

$$\alpha_L \geqslant \frac{C\tau p'dr}{32} = \Omega \left( C \left( (1 + \ln p'dNT)^3 (1 + \ln \kappa) \right) p'dr \right) \,.$$

For a choice that verifies

$$C = \Theta \left( \left( 1 + \ln \frac{1}{\delta} \right)^3 \right) \,, \tag{32}$$

we have the following lower bound for $\alpha_L$:

$$\frac{\alpha_L}{4\gamma'^2} = \Omega \left( (1 + \ln p'dNT)(1 + \ln \kappa) \left( 1 + \ln \frac{1}{\delta} \right) p'dr \right) \,. \tag{33}$$

The condition Equation (31) is satisfied if the following three conditions are individually satisfied:

$$\textbf{(i)} \quad \frac{\alpha_L}{4\gamma'^2} \geqslant \ln \left( \ln \left( \frac{\gamma'\alpha_U}{\alpha_L} \right) + 1 \right) \,,$$

$$\textbf{(ii)} \quad \frac{\alpha_L}{4\gamma'^2} \geqslant 9p'dr \ln \frac{Dp'}{\epsilon} \,,$$

$$\textbf{(iii)} \quad \frac{\alpha_L}{4\gamma'^2} \geqslant \ln \frac{4}{\delta} \,,$$

as the left-hand side of Equation (31) is increasing in $C$, while the right-hand side is decreasing in $C$, we can pick the maximum $C$ that satisfies all three conditions.

**Condition (i).** By Theorem C.11, we have

$$2D\|L_\star\|_{\mathrm{op}} \geqslant (D+1)\|L_\star\|_{\mathrm{op}} \geqslant (D+1)\sigma_{\min}(L_\star) \geqslant 1 \,,$$

and

$$\frac{\alpha_U}{\alpha_L} \leqslant \frac{2D\|L_\star\|_{\mathrm{op}}\alpha_U}{\alpha_L} = \mathcal{O} \left( \sqrt{NT}\kappa^2 \right) \,.$$

As $x \geqslant \ln x + 1$ for any $x \geqslant 0$, we have

$$\ln \left( \ln \left( \frac{\gamma'\alpha_U}{\alpha_L} \right) + 1 \right) \leqslant \ln \frac{\gamma'\alpha_U}{\alpha_L} = \mathcal{O} \left( \ln 2dNT + \ln \kappa + \ln \frac{1}{\delta} \right) \,.$$

The lower bound in Equation (33) is sufficient to satisfy this condition.

**Condition (ii).** We have that

$$
\begin{aligned}
\ln \frac{Dp'}{\epsilon} &\leqslant \ln Dp' \max \left\{ \sqrt{\frac{\epsilon_1(\delta/4)}{\alpha_L}}, \frac{4\epsilon_2(\delta/4)}{\alpha_L}, \frac{1}{D} \right\} \\
&\leqslant \ln Dp' \max \left\{ \frac{D\epsilon_1(\delta/4)}{\alpha_L}, \frac{4\epsilon_2(\delta/4)}{\alpha_L}, \frac{1}{D} \right\} \\
&= \mathcal{O} \left( \ln p' NT + \ln \kappa + \ln \frac{1}{\delta} \right) .
\end{aligned}
$$

The lower bound in Equation (33) is sufficient to satisfy this condition.

**Condition (iii).** This condition is readily verified by Equation (33).

**Remark C.27.** *The choice of $C$ in Equation (32) and $\tau$, which appears in the bounds, have third-order logarithmic dependencies on $\delta$ and the problem parameters, respectively. This arises because the error is bounded above by second-order terms in $\delta$ and problem parameters. An additional logarithmic factor appears due to the discretization of the problem. Thus, assuming bounded noise, i.e., $\sup \|\xi_t^{(n)}\|_\infty \leqslant B$ for some $B > 0$, one can replace the third-order logarithmic dependencies with first-order counterparts, along with dependency on $B$.*

# D APPROXIMATE EMPIRICAL RISK MINIMIZATION

In this section, we extend the results of Section C to approximate empirical risk minimizers $\tilde{A}$ as in Equation (12). We begin with trivial extensions of Theorem C.14 and Theorem C.15 to approximate minimizers.

**Proposition D.1.** *Let $\tilde{A}$ be an estimate that provides an $\epsilon_{\mathrm{tr}}$-approximation to the loss of $\mathcal{L}(A_{p'}^\star)$:*

$$
\mathcal{L}(\tilde{A}) \leqslant \mathcal{L}(A_{p'}^\star) + \epsilon_{\mathrm{tr}} .
$$

*In particular, one can choose $\tilde{A}$ as an $\epsilon_{\mathrm{tr}}$-approximate empirical risk minimizer, i.e.,*

$$
\tilde{A} \in \left\{ A \in \mathcal{A}(\mathcal{D}) \mid \mathcal{L}(A) \leqslant \mathcal{L}(\hat{A}) + \epsilon_{\mathrm{tr}} \right\} ,
$$

*where $\hat{A}$ is the empirical risk minimizer defined in Equation (21). Then, $\tilde{A}$ satisfies*

$$
\left\| \Delta_{\tilde{A}} L_\star E \right\|_F^2 \leqslant 2 \operatorname{Tr} \left( E^\top L_\star^\top \Delta_{\tilde{A}}^\top \left( I - M_{A_{p'}^\star} \right) L_\star E \right) + NT\epsilon_{\mathrm{tr}} .
$$

*Proof.* The proof follows from Equation (22). $\square$

**Corollary D.2.** *Observe that for $p' = p$, Theorem D.1 reads*

$$
\left\| \Delta_{\tilde{A}} L_\star E \right\|_F^2 \leqslant 2 \operatorname{Tr} \left( E^\top \Delta_{\tilde{A}} L_\star E \right) + NT\epsilon_{\mathrm{tr}} .
$$

*For $p' < p$, one can write the following relaxed condition for any $\tilde{A}$:*

$$
\left\| \Delta_{\tilde{A}} L_\star E \right\|_F^2 \leqslant 2\eta \operatorname{Tr} \left( E^\top \Delta_{\tilde{A}} L_\star E \right) + NT\epsilon_{\mathrm{tr}} .
$$

*Proof.* The proof follows from Theorem C.15. $\square$

We divide the analysis into two cases: $2\eta \operatorname{Tr} \left( E^\top \Delta_{\tilde{A}} L_\star E \right) > NT\epsilon_{\mathrm{tr}}$ and $2\eta \operatorname{Tr} \left( E^\top \Delta_{\tilde{A}} L_\star E \right) \leqslant NT\epsilon_{\mathrm{tr}}$. In the first case, we can directly apply the results of Section C to approximate minimizers after noting that Theorem C.15 implies that

$$
\left\| \Delta_{\tilde{A}} L_\star E \right\|_F^2 \leqslant 2 \, (2\eta) \operatorname{Tr} \left( E^\top \Delta_{\tilde{A}} L_\star E \right) .
$$

This is equivalent to doubling the constant $\eta$ in the main theorem and does not change our results.

In the second case, we have the following:

$$\left\|\Delta_{\tilde{A}}L_\star E\right\|_F^2 \leqslant 2NT\epsilon_{\text{tr}}\,.$$

Recall the lower isometry proven in Equation (26):

$$\inf_{A \in \mathcal{A}(D)\backslash\mathcal{G}(C,D)} \|\Delta_A L_\star E\|_F^2 \geqslant \frac{\sigma^2}{8}\sigma_{\min}(L_\star)^2 CD^2\eta^2\tau p'dr\,.$$

If $\epsilon_{\text{tr}}$ is such that

$$2NT\epsilon_{\text{tr}} \leqslant \frac{\sigma^4}{8}\sigma_{\min}(L_\star)^2 CD^2\eta^2\tau p'dr\,,$$

then the approximate minimizer $\tilde{A}$ is guaranteed to be in the set $\mathcal{G}(C,D)$ with high probability. By Theorem C.13, $\mathcal{G}(C,D) \subset \mathcal{S}(C,D)$, and thus, we have the desired result $\tilde{A} \in \mathcal{S}(C,D)$.

## E    EXPERIMENTS

All experiments in this section are implemented with Python 3 (Van Rossum & Drake, 2009) under PSF license and PyTorch (Paszke et al., 2019) under BSD-3-Clause license. In addition, we use NumPy (Harris et al., 2020) under BSD license.

For all the experiments, $A^\star$ is generated as follows. First, $p$ orthogonal matrices of shape $d \times d$ are sampled. These are then scaled down by $\alpha \cdot p$ where $\alpha$ is arbitrarily set to 0.5. In cases where $A$ needs to be initialized, we use the same recipe for the student model with $p'$ instead of $p$ and set $\alpha = 1$. For experiments with low-rank ground truth, we set arbitrary $d - r$ singular values to 0 following a SVD decomposition. Each experiment in this section has been run over 3 independent seeds and the average is plotted. As the variance is small and the plots usually overlap, we opt to not plot it for visual clarity.

Theorems 4.1 to 4.3 provide rates on estimation error for empirical minimizers. In the following, we study these rates empirically for various values of $p', p, d, N, T$ and $r$ where $r = d$ for full-rank experiments or $r = 5$ for low-rank experiments and $p' = p$ except it is stated otherwise. We use two quantities, $\beta = NT$, the number of total tokens, $\gamma = pdr$, the number of parameters to estimate, to summarize information in the plots. For Theorems 4.1 and 4.3, $\hat{A}$ is computed with the OLS estimator and for Theorem 4.2, $\hat{A}$ is learned with gradient descent with learning rate $\alpha$ on the group-norm regularized loss in Equation (9). The parameter $\lambda$ and learning rate $\alpha$ are tuned by a grid search.

Figure 1 plots the estimation error for $d \in \{5, 10, 15\}, p \in \{5, 10, 15\}, N \in \{1, 5, 10\}$ and $T \in \{1, 5, 10, 25, 50\} \times pdr/N$. The upper bound in Theorem 4.1 scales with the ratio $\beta/\gamma$ up to logarithmic terms as empirically verified by Figure 1. In Figure 2, we verify that there is no individual trend to $p$ and $d$, which implies that the error depends only on $\gamma$. Furthermore, we show the trend in $N$ can be accounted for by incorporating the logarithmic term into $\beta$ to obtain $\tilde{\beta} = \beta/\ln(1 + \sqrt{N})$.

Figure 3 plots the estimation error for different degrees of misspecification where the context length is fixed to $p = 15$. The curves for various $p' \in \{5, 10, 15\}$ overlap, which verify the rate $\gamma/\beta = p'd^2/NT$ predicted by Theorem 4.3 holds.

Figure 4 repeats the same plots for low-rank experiments where $d = 15, r = 5$ are fixed and $p, N$ and $T$ are varied as before. Good estimation of $A$ is not straightforward as $\lambda$ has to be appropriately tuned. Yet, we see that the group-nuclear norm regularized estimators found with gradient descent after tuning on regularization problem $\lambda \in \{10^{-1}, 10^{-2}, 10^{-3}, 10^{-4}, 10^{-5}, 10^{-6}, 10^{-7}\}$ and learning rate $\alpha \in \{10^{-1}, 10^{-2}, 10^{-3}\}$ obtain improved estimation errors than non-regularized OLS estimator. Particularly, the sample efficiency benefits of the group-nuclear norm regularization are amplified in the low-data regime. We leave the analysis of group-nuclear norm regularization as a future work.

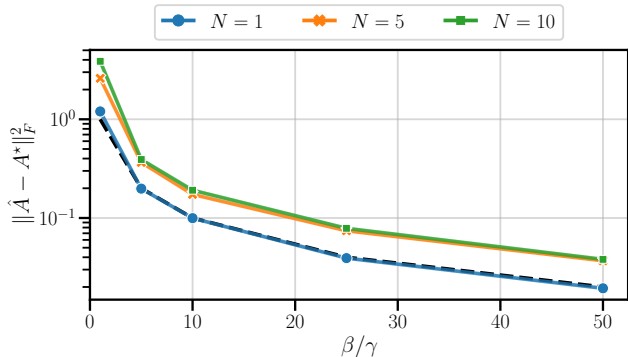

Figure 1: Scaling of estimation error with respect to the ratio $\beta/\gamma = NT/pd^2$ with the OLS estimator. The black dashed line marks the reference value $\gamma/\beta$.

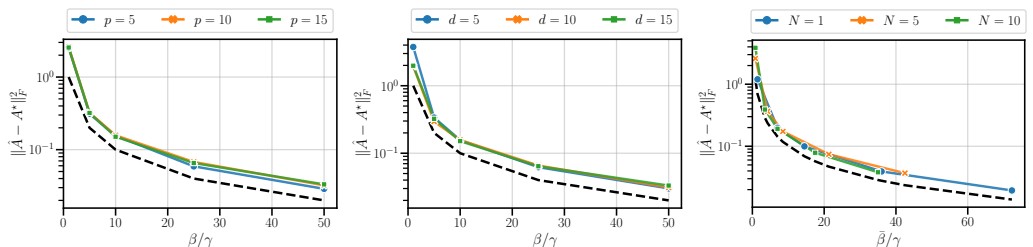

Figure 2: Scaling of estimation error for different values of $p$, $d$ and $N$ with the OLS estimator. Recall that $\beta = NT, \gamma = pd^2$ and $\bar{\beta} = \beta/\ln(1 + \sqrt{N})$. The black dashed lines mark the reference values corresponding to $\gamma/\beta$ and $\gamma/\bar{\beta}$.

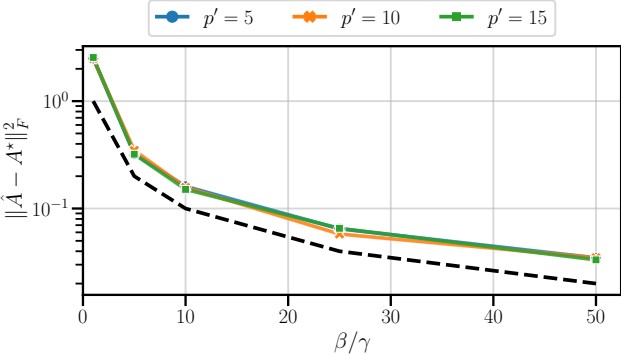

Figure 3: Scaling of estimation error with respect to the ratio $\beta/\gamma = NT/p'd^2$ for different $p' = 5, 10, 15$ with the OLS estimator. The black dashed line marks the reference value $\gamma/\beta$.

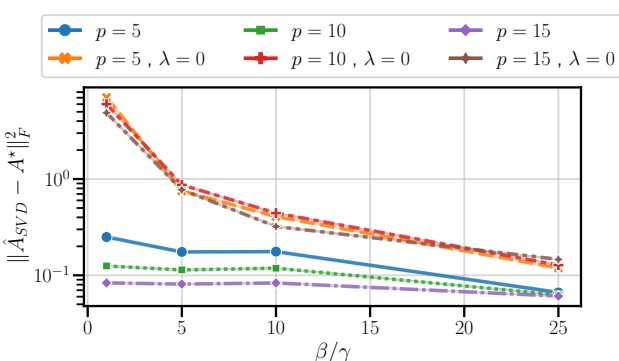

Figure 4: Scaling of estimation error with respect to $\beta/\gamma = \frac{NT}{pdr}$ for different context windows $p = 5, 10, 15$ with the OLS estimator $(\lambda = 0)$ and group-nuclear norm regularized estimators.

