# OpenReview forum: "Long-Context Linear System Identification"
_ICLR.cc/2025/Conference — ICLR 2025 Poster_

### Official Review · Reviewer_o3yE · 2024-10-30

**Soundness:** 2
**Presentation:** 2
**Contribution:** 2
**Rating:** 3
**Confidence:** 4

**Summary:**

The paper provides a low rank approach for long-context linear system identification.

**Strengths:**

The paper provides a low rank approach for long-context linear system identification.

**Weaknesses:**

It's unclear how to tune the rank when we don't know it's true value (e.g. real dataset).

**Questions:**

How do you select the proper rank when it's not known?

---

### Official Review · Reviewer_tdsH · 2024-10-30

**Soundness:** 3
**Presentation:** 2
**Contribution:** 3
**Rating:** 6
**Confidence:** 3

**Summary:**

The article focusses on the problem of identifying the $A_k$ matrices for $k=1,\ldots,p$ where $p$ is the context length when the data is being generated via

$$ x_t=\sum_{k=1}^p A_k^* x_{t-k}+\xi_t $$

where $A_1^*,\ldots,A^*_p$ are $d\times d$ matrices and $\xi_t$ is i.id. noise. The article discusses three problems

$(1)$ Minimizing the empirical loss function under an induced-norm bound on the $A^*$ matrices where the loss function is given by

$$\ell({\bf A})=\frac{1}{NT}\sum_{n=1}^N\sum_{t=p}^T\left\| x_t^{(n)}-\sum_{k=1}^p A_k x_{t-k}^{(n)}\right\|.$$

Here, $T$ is the length of the trajectory and $N$ trajectories are collected.

$(2)$ Minimizing the empiricial loss function with induced-norm constraints on the $A_k$ with an added rank constraint on $A_k$

$(3)$ The same loss function is minimized with an induced norm contraints on $A_k$ matrices and a bound that captures a context length $p'$ which is smaller than the actual context length $p.$

For all the three problems above the article provides non-asymptotic bounds on the Frobenius norm of the error of the estimates with respect to $A_k^*$  that will result from an optimal solution to the problems (1), (2) and (3). The authors provide a discussion that why standard approaches of lifting the state will face challenges, and provide reasons on why the bounds they have obtained are independent of the time taken by the Markov Chain to reach any steady state distribution. The authors further comment on stability conditions.

**Strengths:**

The results provide non-asymptotic bounds on three problems that are well motivated; earlier works have not covered the case where the process has dependency on the past with a context length.

**Weaknesses:**

(1) The non-asymptotic bounds are not with respect to any specific algorithm that takes data and solves the related optimization problems. The authors indicate possible approaches for solving these problems but do not analyze any specific algorithm; however, it stands to reason that the sample complexity will depend on the approach being taken. The rank constrained problem is a particularly challenging one as its not a convex problem.  The authors assume a an optimal solution to the problems. The authors need to comment on whether advances with respect to other works are also in the same spirit or if they analyze known solutions to the optimization problems. Proper justification of the utility of the results need to be provided if the article provides analysis assuming the existence of the optimal solution.

(2) The mathematics is presented in a dense manner; here the approach and the problem description can be better presented  and explained.  In the Sketch of the Proof, some of the matrices such as $E$ are not defined in the main body (its defined in the Appendix; however, the main body should be self-contained; the definitions are buried deep in the Appendix). Some suggestions are to show the matrix operations in more detail to help a reader along.

**Questions:**

(1) The authors discuss how their results do not depend on the mixing time of the Markov Chains involved. The authors can provide better intuition on how they need not consider mixing time in the non-asymptotic bounds obtained.

(2) Can the authors provide more details on the simulations reported. The problems being considered do not  admit closed-form solutions and include non-convex problems and thus should be difficult to solve. How are these challenges reflected in the simulations section.

---

### Official Review · Reviewer_Mair · 2024-11-03

**Soundness:** 3
**Presentation:** 4
**Contribution:** 3
**Rating:** 8
**Confidence:** 3

**Summary:**

The paper considers the linear system identification problem under a long context framework. More specifically, the paper presents:

1. A main result on a theoretical guarantee of the constrained least squares estimator under mild assumptions on the design matrix and sub-gaussianity of noise. This result is shown to parallel previously existing results in the i.i.d. setting under some additional logarithmic factors.
2. An extension of the main result to a low rank setting, showing an improved statistical rate depending on the rank constraint.
3. A further extension of the main result to the case of misspecified context windows, suggesting partial learning occurs for misspecified models.

**Strengths:**

While the topic of linear identification is certainly not new, the theoretical results developed in this paper are novel. The authors clearly stated problem formulations, main results and motivations. Overall, the paper was well written with an enjoyable read.

**Weaknesses:**

There are several minor questions and suggestions regarding confusions in the main text (these are deferred to the questions section below). Experiments were minimal and only provided in the appendix.

**Questions:**

Some minor questions and suggestions include:

1. The first question is about the claim in line 379 that "Importantly, the constant $C$ and the logarithmic terms are independent of the mixing related quantity $\text{max}(1/1-\rho,p).$ Here, $\rho$ is the operator norm of $M_{\textbf{A}^\star}$". However, as stated in line 263, the explicit constant $C(\delta)$ depends on the diameter $D$ which is one of the constraints listed in Assumption 3.4 where it is assumed that the operator norm of $M_{\textbf{A}^\star}$ is less than or equal to $D$. These two statements seem to be contradictory. Can you elaborate on these dependencies?

2. In Equation (14), the variable $E$ is not clearly defined in the main text although it is available in the appendix. It may be helpful to add it in the main text to prevent confusion.

3. The last inequality in Equation (18) appears to be a typo.

4. There are multiple typos in the appendix, and it would be helpful to do a careful revision of the text. For example, line 1184 formatting, line 1287 "martices", line 1378 "rearrainging", to name a few.

5. Considering the main results depend strongly on the condition number of $L_\star$, it would be helpful to include discussions about how this condition number typically behaves. For example, how does this condition number relate to the condition number or singular values of a matrix $A$ if $A_1^\star = A_2^\star = \ldots = A$? How does it behave if $A_i^\star$ have elements sampled i.i.d. from normal distributions? Does the condition number of $L_\star$ also depend on $T$?

6. While in Equation (8) it is stated that the result depends on polylog$(\kappa)$, can you elaborate on whether this result depends on $\log (\kappa)$, or is it actually dependent on O($\kappa$) or other polynomials of $\kappa$? It is not immediately obvious from the proof, however, in many contexts theoretical guarantees of estimators are linearly related to logs of condition numbers. Since $\kappa$ is already the log of the condition number, I think it makes sense to clarify this dependency.

**Details Of Ethics Concerns:**

No ethics concerns were found.

---

### Official Review · Reviewer_RGko · 2024-11-03

**Soundness:** 3
**Presentation:** 3
**Contribution:** 3
**Rating:** 8
**Confidence:** 4

**Summary:**

The authors study the problem of identifying long-context linear systems where the state at any given time depends on a sequence of previous states over an extended context window. In contrast to traditional linear system identification that typically assumes first-order dependencies, this paper focuses on autoregressive processes of order
$p>1$. The authors establish sample complexity bounds, demonstrating a "learning-without-mixing"-type of result. In particular, they show that a slow mixing does not inflate their learning rates. In addition, the authors further extend their results to the setting where the long-context linear model admits a low-rank representations. They also explore the implications of context length misspecification.

**Strengths:**

**Clarity of exposition:** The paper is well-written and well-organized and systematically introduces the problem setting, contributions, and theoretical derivations. Definitions and assumptions are clearly stated, and the logical progression through each theoretical component makes the paper easy to follow.

**Intuitive and well-discussed results:** The concept of "learning-without-mixing" is well-motivated by the authors. This result aligns with the literature on "learning-without-mixing" for linear system. In particular, the authors show that for long-context linear system identification, where long contexts naturally entails a strong sample dependency, it does not necessarily inflate the bounds. Moreover, the low-rank representation learning setting and misspecification scenarios are well-explained, with clear justifications for how each condition affects the error bounds.

**Theoretical contribution:** The theoretical contributions are significant, providing error bounds that extend classical linear system identification results to long-context models, and the learning rates aligns with the literature of learning-without-mixing for linear systems.

**Weaknesses:**

**Misspecification Results and Assumptions:** Section 3.4, particularly Assumption 3.9, imposes a constraint on the misspecified model that may be too restrictive for practical applications. The requirement that $|| (MA^\star - MA^\star_{1:p'})L^\star ||_{\text{op}} \leq D'$ implies that misspecification must remain controlled to a certain degree. The authors could discuss the limitations of Assumption 3.9 if this assumption does not hold in practical settings or offer heuristics for relaxing this constraint.

**Questions:**

1) In Section 5, the authors emphasize that the sample complexity bounds derived remain unaffected by mixing times, highlighting the "learning-without-mixing" result. Could the authors discuss more the slow mixing setting (i.e., when the system is marginally stable), would the learning rates deteriorate?

2) The misspecification results in Section 4 suggest that shorter context lengths can still capture useful structure in long-context systems. Could the authors provide insights into specific applications where such misspecified models are particularly advantageous?

3) I am curious about how coordinate descent minimization could be used to learn $P^\star$ in polynomial time for this setting of long-context linear system identification and the implications of non-isotropic data when updating $P$.

**Minor**: The abbreviation for Ordinary Least Squares (OLS) is used early but is only formally defined in Section 3.2.

---

### Meta-Review · Area_Chair_39A8 · 2024-12-17

**Metareview:**

This paper addresses the problem of linear system identification within a long-context framework. Specifically, it makes the following contributions:

1. Theoretical Guarantee for the Constrained Least Squares Estimator: Under mild assumptions on the design matrix and sub-Gaussian noise, the paper establishes a theoretical guarantee for the constrained least squares estimator. This result parallels existing results in the i.i.d. setting, with an additional logarithmic factor.
2. Extension to the Low-Rank Setting: The main result is extended to a low-rank setting, demonstrating an improved statistical rate that depends on the rank constraint.
3. Misspecified Context Windows: The analysis is further extended to the case of misspecified context windows, revealing that partial learning still occurs under model misspecification.

Most of the reviewers believe that the paper makes substantial contributions, and the AC also agrees with this assessment.

**Additional Comments On Reviewer Discussion:**

Four reviewers have evaluated the paper, and their overall assessment is positive. I agree with their evaluation and believe the paper offers a strong contribution with compelling results.

Two reviewers raised a few technical questions, which the authors have addressed satisfactorily. Another reviewer suggested including remarks on the applicability of the proposed method in the revised version of the paper. I strongly recommend that the authors incorporate these remarks in the final version.

One reviewer recommended a “Reject”; however, their review is very brief and lacks substantive feedback or justification. Given the absence of meaningful critique, I have chosen to disregard this rating in my final assessment of the paper.

---

### Decision · Program_Chairs · 2025-01-22

Accept (Poster)